



# Adding Sea Ice Effects to A Global Operational Model (NEMO v3.6) for Forecasting Total Water Level: Approach and Impact

Pengcheng Wang[1] and Natacha B. Bernier[1]

[1]Recherche en Prévision Numérique Environnementale (RPN-E), Meteorological Research Division (MRD), Environment and Climate Change Canada, Dorval, Quebec, Canada

**Correspondence:** Pengcheng Wang (Pengcheng.Wang@ec.gc.ca)

**Abstract.** In operational flood forecast systems, the effect of sea ice is typically neglected or parameterized solely in terms of ice concentration. In this study, an efficient way of adding ice effects to global total water level prediction systems, via the ice-ocean stress, is described and evaluated. The approach features a novel, consistent representation of the tidal relative ice-ocean velocities based on a transfer function derived from ice and ocean tidal ellipses given by an external ice-ocean model.

The approach and its impact are demonstrated over four ice seasons in the Northern Hemisphere using in-situ observations and model predictions. We show that adding ice effects helps the model reproduce most of the observed seasonal modulations in tides (up to 40% in amplitude and 50° in phase for $M_2$) in the Arctic and Hudson Bay. The dominant driving mechanism for the seasonal modulations is shown to be the under-ice friction, acting in areas of shallow water (less than 100 m), and its accompanied large displacements of amphidromes (up to 125 km). Important contributions from baroclinicity and tide-surge

interaction due to ice-ocean stress are also found in the Arctic. Both mechanisms generally reinforce the seasonal modulations induced by the under-ice friction. In forecast systems that neglect or rely on simple ice concentration parameterizations, storm surges tend to be overestimated. With the inclusion of ice-ocean stress, surfaces stresses are significantly reduced (up to 100% in landfast ice areas). Over the four ice seasons covered by this study, corrections up to 1.0 m to the overestimation of surges are achieved. Remaining limitations regarding the overestimated amphidrome displacements and insufficient ice break-up during

large storms are discussed. Finally, the anticipated trend of increasing risk of coastal flooding in the Arctic, associated with decreasing ice and its profound impact on tides and storm surges, is briefly discussed.

## 1 Introduction

The ice conditions in the Arctic are changing rapidly. As the onset of the ice season is delayed and the return to the ice-free season is advanced (Johnson and Eicken, 2016), the period of exposure to coastal flooding is lengthened. The provision of

accurate and timely forecast of total water level (TWL) in ice-infested waters is thus becoming increasingly important. Under global warming, the increasing effects of the receding ice that protects the shorelines, combined with permafrost thawing that leads to coastal erosion, are resulting in increased exposure to coastal hazards. Many coastal communities in the Arctic and nearby bays and seas are already affected by larger storm surges and rising sea level (Pörtner et al., 2022). For example, Shishmaref, a village on an island off the coast of northern Alaska, is facing the prospect of relocation. Tuktoyaktuk, the major



port of the western Canadian Arctic, is experiencing severe coastal erosion (Whalen et al., 2022), and its shoreline protection structures have been rapidly destroyed by storm surges and accompanying waves.

Sea ice affects both tides and storm surges, the dominant components of TWL, by adjusting the air-sea momentum flux and providing additional friction to the underlying ocean flow. In-situ observations made by tide and bottom pressure gauges have shown remarkable seasonal variability in the $M_2$ tidal amplitude in many parts of the Canadian Arctic, including the

Beaufort Sea and the Amundsen Gulf (up to 50%, Henry and Foreman, 1977; Godin and Barber, 1980), the Kitikmeot Sea (50-60%, Rotermund et al., 2021), and the Hudson Bay (HB) system (8-40%, Prinsenberg, 1988; St-Laurent et al., 2008). Large variability of $M_2$ amplitude was also reported in the Russian Arctic (Kulikov et al., 2018, 2020): up to 63% in the Chukchi Sea (CS) and 9% in the White Sea (Fig. 1). (We note however that the last two amplitude changes are calculated with respect to the annual mean and are thus larger compared to this and other studies that calculate changes using the maxima as the reference.)

Significant delay or advance in the winter $M_2$ phase was also observed: up to 40° in the CS (Kulikov et al., 2018) and eastern HB (Prinsenberg, 1988). On the Ross Ice Shelf of the Antarctic, analyses of Global Positioning System solutions together with tide gauge data (Ray et al., 2021) reveal a counter-intuitive $M_2$ seasonal cycle, associated with suppressed amplitude (%10) and retarded phase during the ice-free season. Recently, altimeter-derived data at high latitudes were also used to study the $M_2$ seasonality for the Arctic and connected regional seas (Bij de Vaate et al., 2021). Although hampered by low temporal

resolution and the presence of ice cover, Bij de Vaate et al. (2021) showed opposing responses to winter ice condition with $M_2$ phase delayed in most of the Arctic, but advanced in the HB.

To understand the underlying physics leading to the seasonal modulation of tides, it is desirable to isolate processes at play. Modelling studies can help separate ice effects from other relevant processes, such as the nonlinear tide-surge interaction (TSI, Bernier and Thompson, 2007) and baroclinicity (Müller et al., 2014). Using a coupled ice-ocean model, St-Laurent

et al. (2008), and later Müller et al. (2014) showed that the observed seasonal $M_2$ modulation in the HB, derived from bottom pressure records, can be largely accounted for by the under-ice friction. As both studies focused on ice processes only, TSI and baroclinicity were not examined. Other studies are based on tide-only models with under-ice friction expressed as additional bottom friction, parameterized solely in terms of ice concentration (e.g., Dunphy et al., 2005; Kleptsova and Pietrzak, 2018) or applied over landfast ice only (Bij de Vaate et al., 2021; Rotermund et al., 2021). These simple methods help produce the

ice-induced modulation of tides over particular regions and periods, but cannot account for its complex spatial and temporal variability.

For storm surges, ice-induced attenuation has been observed in the Baltic Sea (Lisitzin, 1974) and Beaufort Sea (Henry, 1975). Efforts have been made to include such effects in storm surge modelling. Kowalik (1984) and Danard et al. (1989) applied models that include ice-ocean interactions in the Beaufort and Chukchi Seas, but neither of them verified the ice effects

on surges for winter storm events. Zhang and Leppäranta (1995) applied an ice-ocean model in the Baltic Sea and found that the sea surface slope in ice-covered cases may get down to one-third of the ice-free value. More recently, Joyce et al. (2019) incorporated the ice effects on surges through parameterizations of the wind drag coefficient, and showed improvements on the coast of Alaska over particular periods. Kim et al. (2021) adopted the method of Joyce et al. (2019) and showed improvements





for simulated peak winter surges at Tuktoyaktuk. However, such parameterizations depend solely on ice concentration, which
is insufficient to represent the ice strength and its impact on the air-sea momentum flux transfer.

    In Canada, sea ice effects on TWL forecasts are a major concern, particularly in the Canadian Arctic and possibly on the
east coast of Canada. This process is missing in the recently developed global high-resolution (1/12°) TWL system (Wang
et al., 2021, 2022) running operationally at Environment and Climate Change Canada (ECCC). The system is currently under
active development by addressing important physical processes whilst keeping the system easy to maintain and computation-
ally efficient, so that ensemble forecasts can be performed and made available with sufficient lead time to allow maximum
response time for the authorities and the public. Following this principle, we have developed effective and efficient methods
to address TWL contributions from tides, storm surges, baroclinicity and their interactions (Kodaira et al., 2016a, b; Wang
et al., 2021, 2022). In the present study, we attempt to further address sea ice effects following the same principle. A partic-
ular challenge is that simple parameterizations (e.g., based on ice concentration only) are insufficient, whilst coupling with a
sophisticated ice model is not suitable with computational efficiency.

    As in other operational centers, ECCC has recently developed advanced operational ice-ocean systems with data assimila-
tion, and more realistic representations of ice physics and its interaction with the ocean (Smith et al., 2016, 2018; Lemieux
et al., 2015, 2016; Roy et al., 2015). These systems are generally not suitable for accurate and timely water level forecast
as their horizontal resolution is too coarse (1/4°), their computational cost is not sufficiently low, and/or tides or other pro-
cesses critical to TWL forecasts are not considered. However, they can offer information necessary to account for ice effects in
higher resolution, computationally efficient systems optimized to forecast TWLs. In this study, we aim to address the following
questions: (1) Can we design a new parameterization to include ice effects in a global ocean model for forecasting TWL and
improve forecast skill in polar regions? (2) Can we isolate and explain the contribution of dominant physical processes (e.g.,
under-ice friction, baroclinicity, nonlinear tide-surge interaction) to the seasonal modulation of tides?
The structure of the paper is as follows. The observations of coastal TWL are described in Section 2. The ocean model
is introduced in Section 3. The new parameterization of ice effects, via the ice-ocean stress, is described in Section 4. The
experimental design and analysis are presented in Section 5. The impact of adding ice effects on the forecast skill, and the
underlying physics, are examined in Section 6. The results are summarized and discussed in the final section.

## 2   Observations

The present study uses 58 stations grouped into three subregions (Fig. 1). Permanent tide gauges (red circles) in the Arctic
are very sparse and primarily located around the Beaufort Sea, Chukchi Sea and Northern Norway. In an effort to maximize
observations available for verification we collected data, including tide gauge records, bottom pressure records and monthly
tidal constants, from various institutes and publications (Table 1) dating back as early as 1957. Our initial criteria was that
stations have records of at least 12 continuous months. More details about data availability, for each station, are given in Fig. 2.
Data quality control was conducted by removing isolated and clustered spikes in TWL records and tidal residuals following
careful visual inspection. In addition, tide gauge record at station 2 before 1969 was not used as it shows significantly different



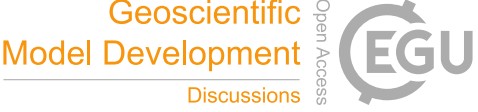

**Table 1.** Summary of water level observations collected from various institutes and publications (see Fig. 1 for station code and Fig. 2 for data availability). Abbreviations are used for Marine Environmental Data Service (MEDS), National Oceanic and Atmospheric Administration (NOAA), University of Hawaii Sea Level Center (UHSLC), and European Marine Observation and Data Network (EMODnet).

| Data type | Data source | Station code |
|---|---|---|
| Hourly tide gauge records | MEDS | 2–12, 31–33, 39, 42–58 |
| | NOAA | 13, 14, 15 |
| | UHSLC | 1, 16, 23–30 |
| | EMODnet | 22 |
| Hourly bottom pressure records | St-Laurent et al. (2008) | 34–38, 40, 41 |
| Monthly tidal constants | Kulikov et al. (2018) | 17–21 |

statistical properties (e.g., variance, seasonal cycle, datum) than the remaining data. Bottom pressure record at station 40 from August 2003 to August 2004 was discarded as it has much coarser temporal resolution than the remainder of the record.

## 3 The ocean model

The NEMO modelling framework (Madec, 2008) is used to solve the governing equations (i.e., the momentum equation, continuity equation and equations for heat and salt transport). They are as follows:

$$\frac{\partial \boldsymbol{u}_h}{\partial t} + \boldsymbol{u} \cdot \nabla \boldsymbol{u}_h + f \times \boldsymbol{u}_h = -\nabla_h [\frac{p_a}{\rho_0} + g(1 - \alpha_s)\eta - g\eta_A$$

$$+ g \int_z^0 \frac{\rho - \rho_0}{\rho_0} dz] + A_h \nabla_h^2 \boldsymbol{u}_h + \frac{\partial}{\partial z}(A_z \frac{\partial \boldsymbol{u}_h}{\partial z}) - \lambda(\boldsymbol{x})\langle \bar{\boldsymbol{u}}_h - \bar{\boldsymbol{u}}_{obs} \rangle, \tag{1}$$

$$\nabla \cdot \boldsymbol{u} = 0, \tag{2}$$

$$\frac{\partial T}{\partial t} + \nabla \cdot (T\boldsymbol{u}) = K_h \nabla_h^2 T + \frac{\partial}{\partial z}(K_z \frac{\partial T}{\partial z}) - r(T - T_f), \tag{3}$$

$$\frac{\partial S}{\partial t} + \nabla \cdot (S\boldsymbol{u}) = K_h \nabla_h^2 S + \frac{\partial}{\partial z}(K_z \frac{\partial S}{\partial z}) - r(S - S_f), \tag{4}$$

where $\boldsymbol{u}_h$ represents the horizontal velocity vector $(u, v)$, $\boldsymbol{u}$ denotes the complete velocity vector in three dimensions $(u, v, w)$, $f$ is the Coriolis parameter, $p_a$ denotes atmospheric pressure at the sea level, $\rho_0$ denotes the reference density (1025 kg m$^{-3}$), $\eta$ denotes the sea surface height, and $\eta_A$ represents the gravitational tidal potential. The depth-dependant coefficient $\alpha_s$ is



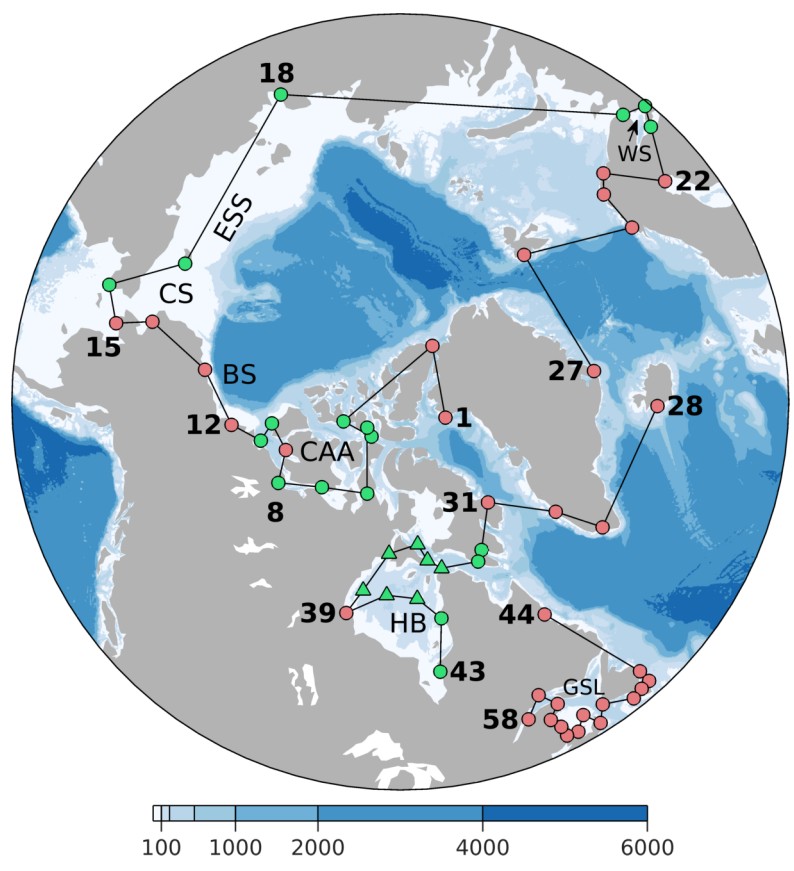

**Figure 1.** Tide gauges (circles) and in-situ moorings (triangles) used in the present study. Red/green symbols indicate that data are available/unavailable during our study period, November 2018 to April 2022 (see Fig. 2 for data availability). The contour map shows bathymetry features in m. Abbreviations are used for the White Sea (WS), East Siberian Sea (ESS), Chukchi Sea (CS), Beaufort Sea (BS), Canadian Arctic Archipelago (CAA), Hudson Bay (HB), and Gulf of St. Lawrence (GSL). The three subregions, their abbreviations and stations numbers, are as follows: (1) Arctic, *Arctic*, 1-27; (2) North Atlantic and Hudson Bay, *NAHB*, 28-43; (3) Northwest Atlantic, *NWA*, 44-58.

used to parameterize the impact of self-attraction and loading (Stepanov and Hughes, 2004). The lateral eddy viscosity and diffusivity coefficients are set to constant ($A_h = 100 \, \mathrm{m^2 \, s^{-1}}$ and $K_h = 10 \, \mathrm{m^2 \, s^{-1}}$), and the vertical eddy viscosity and diffusivity

coefficients ($A_z$ and $K_z$) are determined using the Turbulent Kinetic Energy scheme introduced in Gaspar et al. (1990).

In Eq. 1, the last term on the right-hand side represents the tidal nudging technique introduced in Wang et al. (2021). It nudges the model's depth-averaged current $\bar{\boldsymbol{u}}_h$ towards the observed current $\bar{\boldsymbol{u}}_{obs}$ calculated using the tidal amplitude and phase of eight major tidal constituents ($M_2$, $S_2$, $N_2$, $K_2$, $O_1$, $K_1$, $P_1$ and $Q_1$) provided by TPXO8 (Egbert and Erofeeva, 2002). The angle brackets indicates that the nudging is filtered temporally to isolate variability in tidal frequency bands. The strength

of the nudging is determined by a spatially varying coefficient $\lambda(\boldsymbol{x})$. South of 66°N its global distribution is given by Wang



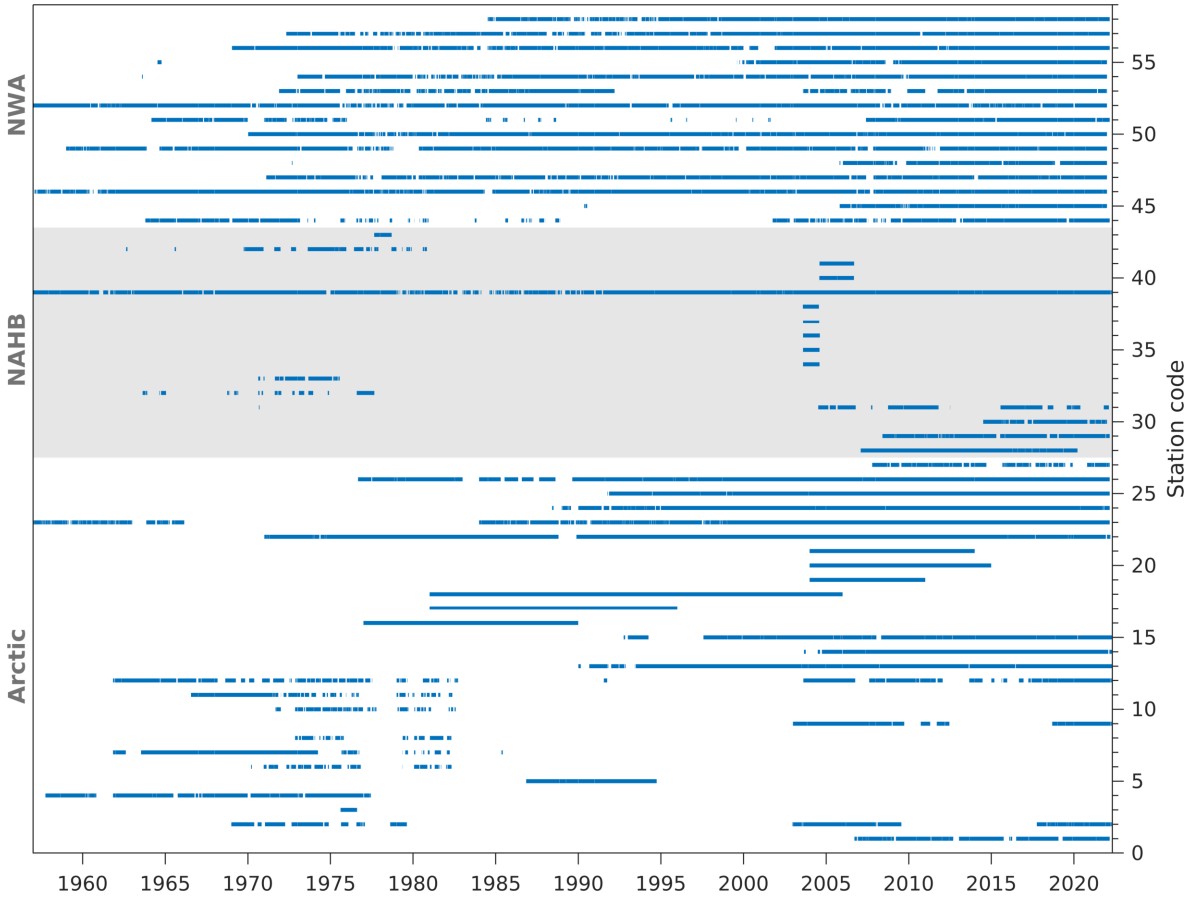

**Figure 2.** Availability of observed water levels as function of station code (see Fig. 1) from January 1957 to April 2022.

et al. (2021). North of 66°N, we set $\lambda(\boldsymbol{x})$ to zero because the nudging would damp the ice-induced seasonal modulation of tides. Recall that TPXO8 does not take ice effects into account and so nudging is not desired when ice effects are considered.

In Eqs. (3) and (4), the last terms on the right-hand side correspond to the nudging of the model's temperature and salinity ($T$ and $S$) towards operational forecasts ($T_f$ and $S_f$) provided by a coarser resolution (1/4°), data-assimilative system, the

Global Ice Ocean Prediction System (GIOPS; Smith et al., 2016, 2018). The strength of the nudging is controlled by a spatially uniform coefficient $r$. We set $r = 0.2 \ \mathrm{d}^{-1}$, and this adds the high-quality low-frequency variability (with periods exceeding about 15 d), provided by the 1/4° model, to our TWL model, while allowing high-frequency variability to evolve freely (Wang et al., 2022).

At the surface, the boundary condition for Eq. (1) is given by

$$A_z \left. \frac{\partial \boldsymbol{u}_h}{\partial z} \right|_{z=0} = \frac{\boldsymbol{\tau}_s}{\rho_0}, \quad \boldsymbol{\tau}_s = \boldsymbol{\tau}_{\mathrm{ao}} = \rho_a C_{\mathrm{ao}} |\boldsymbol{u}_{10}| \boldsymbol{u}_{10} \tag{5}$$





where $\rho_a$ denotes the air density, and $\boldsymbol{u}_{10}$ is the wind velocity at 10 m height. In the absence of sea ice, the surface stress $\boldsymbol{\tau}_s$ equals the air-ocean stress $\boldsymbol{\tau}_{\mathrm{ao}}$. The air-ocean drag coefficient $C_{\mathrm{ao}}$ equals $1.2 \times 10^{-3}$ for $|\boldsymbol{u}_{10}| < 8$ m s$^{-1}$, and then increases linearly with $|\boldsymbol{u}_{10}|$ with a slope of $0.065 \times 10^{-3}$ for every 1 m s$^{-1}$ increase in $|\boldsymbol{u}_{10}|$ (Bernier and Thompson, 2007). Hourly fields of $\boldsymbol{u}_{10}$ and $p_a$ were obtained from the assimilation component of ECCC's operational Global Deterministic Prediction
System (GDPS; Buehner et al., 2015) with a grid spacing of roughly 15 km.

At the bottom, the boundary condition for Eq. (1) is

$$A_z \left. \frac{\partial \boldsymbol{u}_h}{\partial z} \right|_{z=-H} = \frac{\boldsymbol{\tau}_b}{\rho_0}, \quad \boldsymbol{\tau}_b = \rho_0 C_{db} |\boldsymbol{u}_b| \boldsymbol{u}_b \qquad (6)$$

where $\boldsymbol{\tau}_b$ refers to the bottom stress, $\boldsymbol{u}_b$ denotes the current velocity at the bottom, and $C_{db}$ is the bottom drag coefficient which is set equal to $2.5 \times 10^{-3}$.

The extended version of a tri-polar ORCA grid (eORCA12) is used as the model grid. It covers the Antarctic ice shelves and has a horizontal grid spacing of $1/12°$. The bathymetry is obtained from GEBCO_2014 (Weatherall et al., 2015), with local adjustments made in the HB and on the Labrador and Newfoundland Shelf, based on bathymetric data provided by F. Lyard (personal communication). Following Wang et al. (2022), the vertical grid is 9 z-levels, which is able to capture baroclinic variability via $T$ and $S$ nudging whilst maintaining low computational cost. Partial steps are employed for the bottom layer
to achieve a more accurate representation of the bathymetry. Mode-splitting was used with time steps of 240 s and 6 s for the internal and external modes, respectively.

## 4 Surface stress in the presence of sea ice

In this section, we describe the surface stress $\boldsymbol{\tau}_s$ in the presence of sea ice. We review parameterizations or models previously developed and introduce a new, cost-efficient, method for its parameterization in TWL systems.
In the presence of sea ice, $\boldsymbol{\tau}_s$ is generally approximated by a combination of the air-ocean stress $\boldsymbol{\tau}_{\mathrm{ao}}$ (see Eq. (5)) and the ice-ocean stress $\boldsymbol{\tau}_{\mathrm{io}}$ weighted by the ice concentration $\alpha$,

$$\boldsymbol{\tau}_s = (1-\alpha)\boldsymbol{\tau}_{\mathrm{ao}} + \alpha\boldsymbol{\tau}_{\mathrm{io}}. \qquad (7)$$

$\boldsymbol{\tau}_{\mathrm{io}}$ can be parameterized by a quadratic drag law in terms of the relative velocity between ice and surface currents ($\boldsymbol{u}_{\mathrm{ice}} - \boldsymbol{u}_{\mathrm{surf}}$),

$$\boldsymbol{\tau}_{\mathrm{io}} = \rho_0 C_{\mathrm{io}} |\boldsymbol{u}_{\mathrm{ice}} - \boldsymbol{u}_{\mathrm{surf}}| (\boldsymbol{u}_{\mathrm{ice}} - \boldsymbol{u}_{\mathrm{surf}}), \qquad (8)$$

where $C_{\mathrm{io}}$ is the ice-ocean drag coefficient.

To address $\boldsymbol{\tau}_{\mathrm{io}}$, there are several options with different levels of complexity. The most complex option is to couple the ocean model with a sophisticated ice model that simulates ice thermodynamics, dynamics, transport and ridging (Hunke et al., 2010). Unfortunately, such option is not suitable for relatively high resolution systems built with computational efficiency in mind.
A simpler option is to solve the ice momentum equation with prescribed ice concentration and ice thickness. However, the





major time-consuming part in most modern ice modelling, the sub-cycling of the standard elastic-viscous-plastic solver for ice dynamics (Hunke and Dukowicz, 1997), is still required. In addition, the simplified coupling neglects mass transport, and so it has deleterious effects known as 'artificial inertial resonance' in the presence of both tidal and wind forcing (Hibler et al., 2006). Another option is to take $\boldsymbol{u}_{\mathrm{ice}} - \boldsymbol{u}_{\mathrm{surf}}$ directly from an external ice-ocean model, which requires only negligible additional

computational cost. The main issue with this option is the potentially large inconsistency in predicted tides between the TWL and external ice-ocean models. These differences can have various sources such as model resolution, bottom topography, open boundary conditions and parameterization of dissipation. In the following section, we propose and evaluate a new approach to address these inconsistencies.

### 4.1 Mapping ice effects on currents

To address the inconsistency issue in tides associated with the use of an external ice-ocean model to define the ice-ocean stress, while maintaining computational efficiency, we decompose the relative velocity into a tidal component (denoted with the superscript T) and a residual/surge component (superscript S),

$$\boldsymbol{u}_{\mathrm{ice}} - \boldsymbol{u}_{\mathrm{surf}} = (\boldsymbol{u}_{\mathrm{ice}}^{\mathrm{T}} - \boldsymbol{u}_{\mathrm{surf}}^{\mathrm{T}}) + (\boldsymbol{u}_{\mathrm{ice}}^{\mathrm{S}} - \boldsymbol{u}_{\mathrm{surf}}^{\mathrm{S}}). \tag{9}$$

As $\boldsymbol{u}_{\mathrm{ice}}^{\mathrm{T}}$ is mainly forced by $\boldsymbol{u}_{\mathrm{surf}}^{\mathrm{T}}$, we introduce a transfer function, such that,

$\quad \boldsymbol{u}_{\mathrm{ice}}^{\mathrm{T}} \approx a^{\mathrm{T}}(\boldsymbol{x}) \mathbf{R}(\varphi(\boldsymbol{x})) \boldsymbol{u}_{\mathrm{surf}}^{\mathrm{T}}, \tag{10}$

where

$$\mathbf{R}(\varphi(\boldsymbol{x})) = \begin{bmatrix} \cos\varphi(\boldsymbol{x}) & -\sin\varphi(\boldsymbol{x}) \\ \sin\varphi(\boldsymbol{x}) & \cos\varphi(\boldsymbol{x}) \end{bmatrix}, \tag{11}$$

and the spatially-varying scale factor $a^{\mathrm{T}}$ and the rotation angle $\varphi$ can be inferred from $\boldsymbol{u}_{\mathrm{ice}}^{\mathrm{T*}}$ and $\boldsymbol{u}_{\mathrm{surf}}^{\mathrm{T*}}$ provided by external ice-ocean models (the asterisk * denotes quantities from external models). Specifically, $a^{\mathrm{T}}$ and $\varphi$ are derived by scaling and

rotating the ice and ocean tidal ellipses so that their semi-major axes are equal. The tidal relative velocity is thus written as,

$$\boldsymbol{u}_{\mathrm{ice}}^{\mathrm{T}} - \boldsymbol{u}_{\mathrm{surf}}^{\mathrm{T}} = [a^{\mathrm{T}}(\boldsymbol{x}) \mathbf{R}(\varphi(\boldsymbol{x})) - \mathbf{I}] \boldsymbol{u}_{\mathrm{surf}}^{\mathrm{T}}, \tag{12}$$

where $\mathbf{I}$ is the identity matrix.

Unlike periodic tides, storm surges are sporadic and occur on a local scale driven by the atmospheric forcing. Applying a transfer function to $\boldsymbol{u}_{\mathrm{surf}}^{\mathrm{S}}$, similar to Eq. (10), is not feasible, since $\boldsymbol{u}_{\mathrm{surf}}^{\mathrm{S}}$ is forced by $\boldsymbol{u}_{\mathrm{ice}}^{\mathrm{S}}$. Instead, we expect that the incon-

sistency in surges between our model and the external ice-ocean model is acceptable given that their atmospheric forcing are similar. The residual relative velocity can thus be taken directly from the external model, but scaled to account for differences in the valid depths of $\boldsymbol{u}_{\mathrm{surf}}^{\mathrm{S}}$ and $\boldsymbol{u}_{\mathrm{surf}}^{\mathrm{S*}}$,

$$\boldsymbol{u}_{\mathrm{ice}}^{\mathrm{S}} - \boldsymbol{u}_{\mathrm{surf}}^{\mathrm{S}} = a^{\mathrm{S}}(\boldsymbol{u}_{\mathrm{ice}}^{\mathrm{S*}} - \boldsymbol{u}_{\mathrm{surf}}^{\mathrm{S*}}), \tag{13}$$





where $a^S$ is the scale factor that can be optimized based on observations (see Section 4.3 for details).

Finally, the total relative velocity is,

$$\boldsymbol{u}_{\text{ice}} - \boldsymbol{u}_{\text{surf}} = [a^T(\boldsymbol{x})\mathbf{R}(\varphi(\boldsymbol{x})) - \mathbf{I}]\boldsymbol{u}_{\text{surf}}^T + a^S(\boldsymbol{u}_{\text{ice}}^{S*} - \boldsymbol{u}_{\text{surf}}^{S*}). \tag{14}$$

In practice, $\boldsymbol{u}_{\text{surf}}^T = \langle \boldsymbol{u}_{\text{surf}} \rangle$ can be obtained using an efficient online tidal filter (Wang et al., 2021) denoted by the angle brackets. Note that the same filter is also used for the tidal nudging shown in the last term on the right side of Eq. (1).

### 4.2    Ice-ocean stress

Gridded fields of hourly ice concentration ($\alpha$), ice velocity ($\boldsymbol{u}_{\text{ice}}^{T*}$, $\boldsymbol{u}_{\text{ice}}^{S*}$) and surface current ($\boldsymbol{u}_{\text{surf}}^{T*}$, $\boldsymbol{u}_{\text{surf}}^{S*}$) were obtained from the assimilation component of GIOPS (Smith et al., 2016, 2018) developed and run operationally at ECCC. In GIOPS, the CICE-based (Hunke et al., 2010) ice component has 10 categories of ice thickness, and the NEMO-based ocean component has 50 vertical levels and a horizontal resolution of 1/4°. The initialization of GIOPS involves using analyses created by Mercator Océan's System d'Assimilation Mercator version 2 (SAM2, Tranchant et al., 2008). Further information regarding

the initialization procedures can be found in Smith et al. (2018). We note that although GIOPS is a global system, its model grid has a southern limit at about 77°S, which excludes ice cavities in the Ross Sea and Weddell Sea. In the present study, we thus focus on ice-infested waters of the Northern Hemisphere.

As in many global ice-ocean systems, the operational GIOPS does not include tides. To obtain $\boldsymbol{u}_{\text{ice}}^{T*}$ and $\boldsymbol{u}_{\text{surf}}^{T*}$, we reran GIOPS by activating the astronomical tidal potential forcing. In the present study, the transfer function is updated monthly,

which is sufficient to capture its seasonality. We focus on four major tidal constituents ($M_2$, $S_2$, $K_1$, $O_1$) in GIOPS as they can be adequately resolved with monthly harmonic analyses.

Fig. 3 (top panels) illustrates the monthly estimates of $a^T$ for $M_2$ from December 2020 to March 2021. Note that $a^T$ captures the mobility of sea ice: landfast for $a^T = 0$, "non-free" drift for $0 < a^T < 1$, and free drift for $a^T = 1$. In theory, only landfast ice and non-free drift ice exert an friction to the underling ocean flow. Regions of $a^T \to 0$ are seen along the Arctic coast, in parts

of the Canadian Arctic Archipelago (CAA) and the East Siberian Sea (ESS). Identified regions are in reasonable agreement with observed landfast ice occurrences (bottom panels). Non-free drift ice is found further away from the coast in the Arctic and HB, and in particular it covers broad areas of the ESS and CS. We note that not all the identified landfast or non-free drift ice are relevant for the seasonality of tides. We return to this point later (see the end of Section 6.2.2). The rotation angle $\varphi$ is only relevant for drift ice. Its main impact is found in the ESS and CS where the tidal ice and ocean velocity vectors can

be nearly 180° out of phase (not shown), effectively enhancing the under-ice friction. Elsewhere, the absolute value of $\varphi$ is relatively small (within 20°) which has minimal effects on the calculation of the under-ice friction.

Results for the other three constituents are roughly similar to $M_2$ (not shown), although over some regions (e.g., ESS and CS) $K_1$ and $O_1$ are too weak to derive reliable estimates. This similarity between constituents indicates that the transfer function, or the response of sea ice relative to tidal current, is largely determined by ice characteristics. Thus each monthly transfer function

for $M_2$ was applied to other constituents in the present study. We note that the main advantage of this new approach is that it is not sensitive to differences in predicted tides between the ice-ocean and TWL models as the mapping of ice-effects is achieved





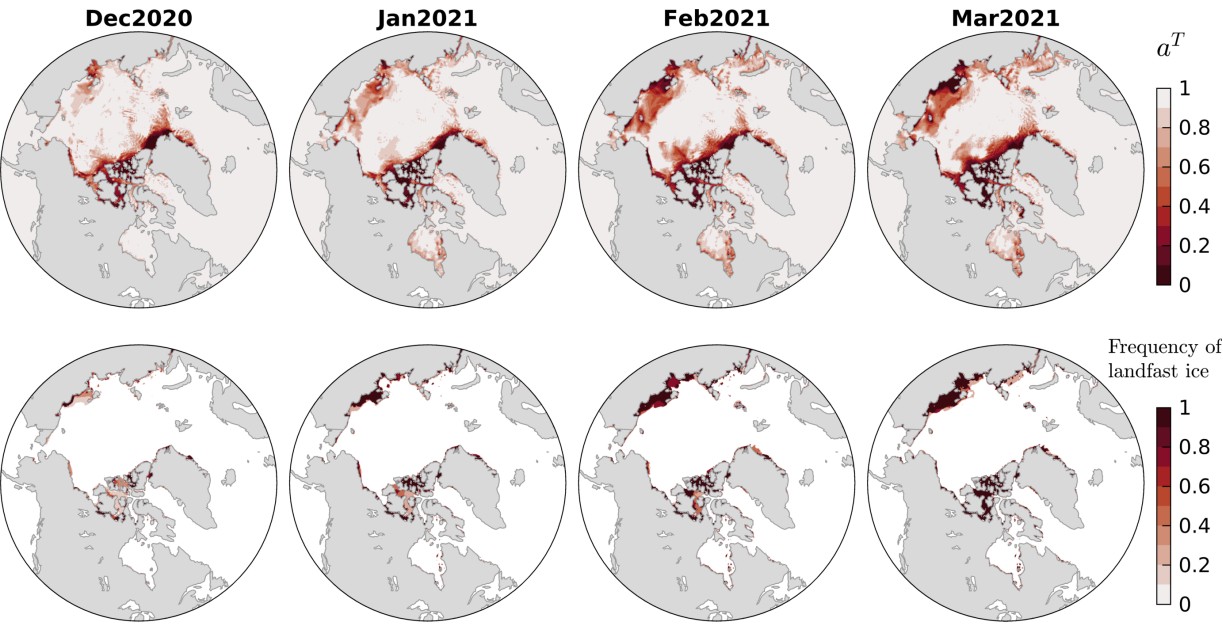

**Figure 3.** Top panels: Derived monthly $a^{\mathrm{T}}$ for the $M_2$ tide for December 2020 to March 2021. Bottom panels: Observed frequency of landfast ice occurrence, obtained from the National Snow and Ice Data Center, for the same period.

via a transfer function. Therefore, in theory, the approach can be used with any ice-ocean systems, regardless of the model skill in tides so long as they are realistic.

### 4.3 Ice-ocean drag coefficient

The parameterization of the ice-ocean drag coefficient $C_{\mathrm{io}}$ is a complex issue, as $C_{\mathrm{io}}$ depends on various ice characteristics such as surface roughness, floe size, ridge height, and keel depth (Lu et al., 2011; Tsamados et al., 2014). Instead, constant values are commonly used and determined by matching model predictions with observations (e.g., St-Laurent et al., 2008; Rotermund et al., 2021). Typical measured $C_{\mathrm{io}}$ values range from $1.05 \times 10^{-3}$ to $4.70 \times 10^{-2}$ in field investigations (see Table 1 in Lu et al., 2011). GIOPS has 50 vertical levels, and its $C_{\mathrm{io}}$ was set to $2.32 \times 10^{-2}$ based on a log layer assumption, using its first

layer currents at 0.5 m and an undersurface sea ice roughness length scale of 0.030 m (Roy et al., 2015). In the TWL system, only 9 vertical levels are used (Wang et al., 2022) and so our first layer currents are valid at 7.5 m. The $C_{\mathrm{io}}$ value for our model is thus expected to be smaller than that used by GIOPS, but the log layer assumption used in GIOPS is not suitable for our first layer. Based on sensitivity tests, we set $C_{\mathrm{io}}$ to $1.00 \times 10^{-2}$ which produces reasonable agreement between observed and predicted seasonal $M_2$ modulations.

The scale factor for residual relative velocity, $a^{\mathrm{S}}$ in Eq. (14), was optimized to 1.64 based on the comparison with observed residuals. The resulting drag coefficient $(a^{\mathrm{S}})^2 C_{\mathrm{io}}$ for the residual stress based on $\boldsymbol{u}_{\mathrm{ice}}^{\mathrm{S*}} - \boldsymbol{u}_{\mathrm{surf}}^{\mathrm{S*}}$ is $2.69 \times 10^{-2}$, indicating a slightly higher roughness length scale of about 0.036 m, compared to 0.030 m used in GIOPS (Roy et al., 2015), both well within the





**Table 2.** Design of the model experiments.

| | Under-ice friction | Tide-surge interaction | | Baroclinicity |
| | (Tidal component of $\tau_{\mathrm{io}}$) | due to $\tau_b$ | due to $\tau_{\mathrm{io}}$ | (Variable $T$, $S$) |
|---|---|---|---|---|
| $\mathrm{Run_{AO}}$ | | ✓ | | ✓ |
| $\mathrm{Run_{AIO}}$ | ✓ | ✓ | ✓ | ✓ |
| Run1 | ✓ | ✓ | ✓ | |
| Run2 | ✓ | ✓ | | |
| Run3 | ✓ | | | |
| Run4 | | | | |

range given in the literature. As an example, McPhee (2008) provides a mean value of 0.049 m with a standard deviation range between 0.016 and 0.146 m based on estimates for a typical multiyear sea ice floe.

## 5 Experimental design and analysis

Two basic runs, $\mathrm{Run_{AO}}$ and $\mathrm{Run_{AIO}}$, were conducted to examine the impact of adding ice effects on predicted water levels. In $\mathrm{Run_{AO}}$, ice effects are not considered and $\boldsymbol{\tau}_s$ equals $\boldsymbol{\tau}_{\mathrm{ao}}$ (see Eq. (5)). In $\mathrm{Run_{AIO}}$, $\boldsymbol{\tau}_s$ is computed as the combination of $\boldsymbol{\tau}_{\mathrm{ao}}$ and $\boldsymbol{\tau}_{\mathrm{io}}$ (see Eq. (7)). In order to quantify the contribution of individual physical processes on the seasonality of tide, four process-oriented runs were also conducted by gradually removing relevant processes from $\mathrm{Run_{AIO}}$: baroclinic effects (by using constant $T$ and $S$, Run1), TSI due to $\boldsymbol{\tau}_{\mathrm{io}}$ (Run2), TSI due to $\boldsymbol{\tau}_b$ (Run3), under-ice friction or tidal component of $\boldsymbol{\tau}_{\mathrm{io}}$ (Run4). The six runs are summarized in Table 2. Note that we chose to remove processes gradually instead of the more traditional removal of a process at a time, because it is not possible to completely isolate the under-ice friction which is a prerequisite for the TSI due to $\boldsymbol{\tau}_{\mathrm{io}}$. Sensitivity studies (not shown) confirm that the impact of the combined removal approach on other processes, that can be isolated (i.e., the TSI due to $\boldsymbol{\tau}_{\mathrm{io}}$ and $\boldsymbol{\tau}_b$), is negligible. Each model run starts on September 21, 2018 and finishes on April 30, 2022. The first 40 days of a run are discarded to allow for model spin up, which is mainly determined by the spin up of the tidal nudging (Wang et al., 2021).

We use the root mean square error (RMSE) to evaluate the model performance for individual stations. To facilitate the comparison with different scales, we also compute the root mean square (RMS) of the observations. Both metrics were calculated for TWL, tides and tidal residuals at each station. Tides were reconstructed based on the eight major tidal constituents used in this study (see Section 3) using the T_TIDE package of Pawlowicz et al. (2002) .

Monthly harmonic analyses of observed and predicted TWL were conducted to examine the ice-induced seasonality of tides. It is noted that for diurnal tides, there is substantial variability in the standard monthly analysis, possibly due in part to the contamination from non-tidal energy (Cartwright and Amin, 1986). To minimize such effect and focus on the seasonal variability, we conducted another set of monthly analyses using a sliding window of 90 days to obtain the estimates for diurnal





tides only. The unresolvable constituent $K_2$ ($P_1$) was inferred from $S_2$ ($K_1$), and the inference parameters including amplitude ratios and phase differences were taken from the yearly analysis. Nodal corrections were performed. Estimates for stations with a signal-to-noise ratio (SNR, see Pawlowicz et al., 2002 for detail) lower than 2 are not used.

For each year, we calculate the normalized amplitude anomaly ($\Delta \tilde{A}_i$) and phase anomaly ($\Delta \phi_i$) relative to their corresponding values in September ($A_{\mathrm{Sept}}$, $\phi_{\mathrm{Sept}}$) when sea ice has the minimum cover in the Arctic,

$$\Delta \tilde{A}_i = \frac{A_i - A_{\mathrm{Sept}}}{A_{\mathrm{Sept}}} \tag{15}$$

$$\Delta \phi_i = \phi_i - \phi_{\mathrm{Sept}}, \tag{16}$$

where the subscript $i$ denotes a particular month. The ice-induced maximum modulation occurs in March ($\Delta \tilde{A}_{\mathrm{Mar}}$, $\phi_{\mathrm{Mar}}$) when sea ice reaches its maximum. We note that the seasonality of tides can be caused by a variety of mechanisms, including astronomical motions, frictional/advective interactions, and climate processes (e.g., baroclinicity, sea ice, river discharge) (Ray,

2022). In the main text, we focus on the seasonality of the dominant $M_2$ constitute, which has negligible astronomical contribution. Other minor constituents, including $S_2$, $K_1$ and $O_1$, are briefly discussed in the Supplementary material.

## 6    Results

### 6.1    Total water level

We first compare the model skill for $\mathrm{Run}_{\mathrm{AO}}$ and $\mathrm{Run}_{\mathrm{AIO}}$ in predicting TWL in terms of RMSE at 34 permanent tide gauges

(top panel of Fig. 4). Improvements from the addition of ice effects are seen in the Arctic, most particularly in the Canadian Arctic (stations 2, 9, 12). The reductions in RMSE are relatively small (i.e., 1.0-3.3 cm). This is expected considering that sea ice matters mostly in winter, in particular, during winter storms which are relatively rare in parts of the Arctic. For example, over the four ice seasons of the study period, there are only two storms at station 2 and no storm at station 9, both of which are located in the CAA. We note however that the impact on peak water levels can be large (up to 1.0 m, see Section 6.3). The

impact at stations in the northern North Atlantic (28–31) and Gulf of St. Lawrence (44–58) is negligible, indicating that the predicted ice is mostly in free drift over these regions. A slight increase in RMSE value of about 4.0 cm is noted at Churchill (station 39) in the HB. This is largely due to the existing bias in predicted tides under ice-free conditions, thus adding the ice-induced modulation increases the bias in the ice season (middle panel of Fig. 4). However, we note that observations at Churchill have possible quality or drift issues as observed tides have undergone large changes since 1998 (Ray, 2016). Finally,

we note that this evaluation is limited to permanent gauges which are very sparse in the Arctic and HB. Next we focus on ice effects on the seasonality of tides at all available stations, and storm surges during large storm events.



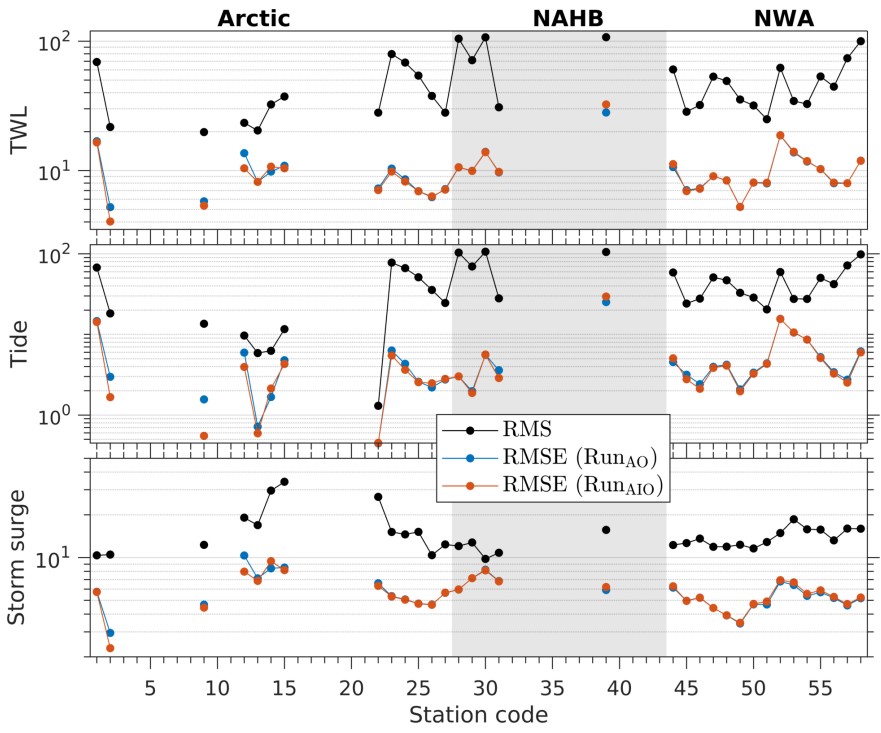

**Figure 4.** RMS of observations (black) and RMSE for Run$_{\text{AO}}$ (blue) and Run$_{\text{AIO}}$ (red) for TWL (top), tides (middle), and storm surges (bottom). All RMS and RMSE values in cm.

## 6.2 Tides

### 6.2.1 Seasonal variability

Figure 5 shows the M$_2$ modulation in March relative to September ($\Delta \tilde{A}_{\text{Mar}}$, $\Delta \phi_{\text{Mar}}$). Comparison of predictions given by

Run$_{\text{AO}}$ and Run$_{\text{AIO}}$ (top two panels) reveals large ice-induced modulations in the Arctic and HB, with the largest modulations occurring around amphidromic points. Adding ice effects generally reduces the amplitudes at the coast (middle-left panel), while the opposite can occur as a result of amphidrome displacement. Adding ice effects also leads to phase delays in most of the Arctic and the CAA, and phase advances in parts of the CAA and most of the HB (middle-right panel). This is due to both the friction-induced decrease in the phase speed of tidal wave and the displacement of amphidromes. We return to this

point later (see Section 6.2.2). The top panels of Fig. 5 also reveal non-negligible M$_2$ modulations in Run$_{\text{AO}}$, indicating that nonlinear TSI and/or baroclinicity also contribute to seasonal variability (see Section 6.2.3 for details).

Observations (filled circles) are also plotted on top of predictions in the top two panels of Fig. 5. Their comparison is further plotted as function of station code in the bottom panel. Adding ice effects significantly improves the model skill in predicting $\Delta \tilde{A}_{\text{Mar}}$ (up to 40%) and $\Delta \phi_{\text{Mar}}$ (up to 50°) at most stations in the Arctic and HB. In the Arctic, improvements are also found

for other smaller constituents (S$_2$, K$_1$, and O$_1$; see Supplementary material for additional details). For M$_2$, one exception is



station 18, where observations show an "anomalous" positive $\Delta\tilde{A}_{\text{Mar}}$ (bottom-left panel of Fig. 5) likely associated with the displacement of local amphidromes. $\text{Run}_{\text{AO}}$ underestimates $\Delta\tilde{A}_{\text{Mar}}$, while $\text{Run}_{\text{AIO}}$ generates a negative $\Delta\tilde{A}_{\text{Mar}}$. To explain this discrepancy, we note that station 18 is located in a relatively small but complex region where multiple small amphidromes are present (see left panel of Fig. 9), and this poses a great challenge for the model to precisely resolve these amphidromes and

their displacements.

Figures 6 and 7 show monthly time series of $\Delta\tilde{A}_i$ and $\Delta\phi_i$ at 14 stations. Selected stations cover various geographical areas where noticeable amplitude or phase modulations are observed, and they all have an $M_2$ amplitude of at least 9 cm and a phase modulation of at least $5°$. Observations show large amplitude reductions (up to 40–50%, Fig. 6) in the Canadian Arctic (stations 6, 7, 12), CS (stations 14, 17), HB (stations 42, 43) and Northumberland Strait (station 53), and large phase modulations (up

to 40–50°, Fig. 7) at Tuktoyaktuk (station 12), in the CS (stations 14, 17) and the eastern HB (station 42). Large modulations can last up to 8 months of the year (e.g., station 12). Model results show that in the absence of ice-induced stress ($\text{Run}_{\text{AO}}$), the predicted $\Delta\tilde{A}_i$ and $\Delta\phi_i$ are pretty flat, except in the CS (stations 14, 17) where other processes (e.g., TSI, baroclinicity, see Section 6.2.3 for details) contribute up to 20% to the amplitude modulation. When ice-induced stress is included ($\text{Run}_{\text{AIO}}$), forecasts of $\Delta\tilde{A}_i$ and $\Delta\phi_i$ are greatly improved at most stations, across the season.

We note that there remains room for improvement. For example, $\Delta\tilde{A}_i$ in $\text{Run}_{\text{AIO}}$ (Fig. 6) shows slight overestimations ($\leq 10\%$) at Resolute and Churchill (stations 4 and 39), moderate underestimations (20%) at Tuktoyaktuk and Inukjuak (stations 12 and 42), and large underestimations (40%) at Shediac Bay (station 53). We note that Shediac Bay is located near the narrow Northumberland Strait (width of about 13 km) that the $1/4°$ external model cannot resolve. For $\Delta\phi_i$ (Fig. 7), the results show moderate (10–20°) underestimation at Tuktoyaktuk, Wrangle, and La Grande Rivière (stations and 12, 17, 43) and

overestimation at Red Dock dock (station 14). Some of these discrepancies can be explained by the over-predicted amphidrome displacements. We return to this point later (see Section 6.2.2).

We next examine the interannual variability in the seasonal $M_2$ modulation at five permanent gauges during the four ice seasons of the study period (Fig. 8). Observations show interannual variability in both the duration and magnitude of the maximum modulation: changes in duration are up to two months (e.g., station 14), while changes in magnitude are up to 10%

in amplitude and 10° in phase. These features are apparently missed by $\text{Run}_{\text{AO}}$, while they are reasonably captured by $\text{Run}_{\text{AIO}}$. One exception is Churchill (station 39) where observations show almost no modulations in amplitude, while $\text{Run}_{\text{AIO}}$ generates 5–10% modulations. We note, as before, that observed tides at Churchill may have quality or drift issues (Ray, 2016).

### 6.2.2 Ice-induced amphidrome displacement

Ice-induced tidal modulations are associated with displacements of amphidromes. In this section, we examine these displace-

ments. We focus on March, at the peak of the ice cover, and examine the $M_2$ amphidrome displacements averaged over the study period (Fig. 9). We note that interannual variabilities in displacements are relatively small except for several small amphidromes in the CS. In general, facing the direction of the displacement, amplitudes decrease in the front whilst they increase in the back. Still facing the direction of the displacement, phase delays and phase advances occur on the left and right sides, respectively. The largest displacement is found in the ESS where the amphidrome (marked "A") moves towards the coast by

**Figure 5.** Modulation of the $M_2$ amplitude ($\Delta \tilde{A}_{\mathrm{Mar}}$, left panels) and phase ($\Delta \phi_{\mathrm{Mar}}$, right panels) in March relative to September. Top two panels: contour map shows results predicted by $\mathrm{Run_{AO}}$ (top panels) and $\mathrm{Run_{AIO}}$ (middle panels) averaged over 2019–2021. Filled circles show results taken from available observations during 1957–2021. Bottom panel: comparison between observation and prediction as a function of station code (Fig. 1). Shaded areas indicate the 10-90 percentile range. Only stations with SNR greater than 2 are plotted.

90–125 km (note arrows in Fig. 5 do not scale with the background distances) due to the ice-induced strong tidal dissipation on
the onshore side of the system (see middle-left panel of Fig. 5). The second largest occurs close to the center of the Arctic where



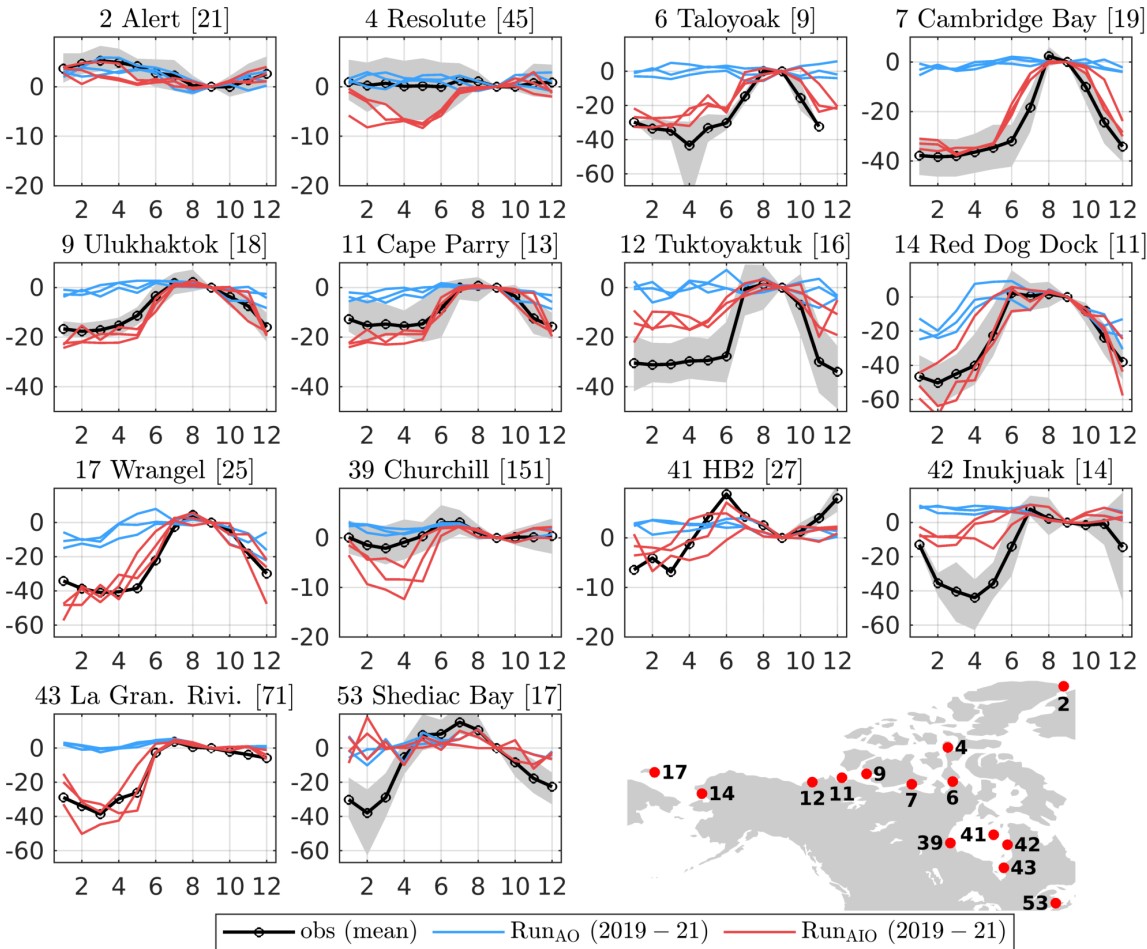

**Figure 6.** Normalized monthly $M_2$ amplitude anomaly ($\Delta \tilde{A}_i$, %) relative to September at 14 selected stations. Observed mean (black line) and 10-90 percentile range (gray shading) are presented based on available data from 1957-2022. Model predictions from 2019-22 are provided by $Run_{AO}$ (blue lines) and $Run_{AIO}$ (red lines). The title of each subplots gives the station code, station name and averaged $M_2$ amplitude in cm in September (number in square brackets). The locations of selected stations and their numbering are shown in the bottom right panel.

the system (marked "B") moves towards the Canadian Arctic by 70–90 km. The two displacements are clearly responsible for the dominant large-scale features of $M_2$ modulations in the Arctic (see Fig. 5). There are also small to moderate displacements (10–50 km) of numerous systems in the Russian Arctic and across the Bering Strait, which affect regional, small-scale modulations.

In the CAA and HB, the frictional effects in Taylor's problem of reflection of Kelvin waves in semi-enclosed basins (Taylor, 1922) explain most of the displacements. The primary effect is the exponential decay of Kelvin wave amplitude in the direction of wave propagation, which causes the amphidromes to shift towards the coast where the reflected Kelvin waves travel (Rie-



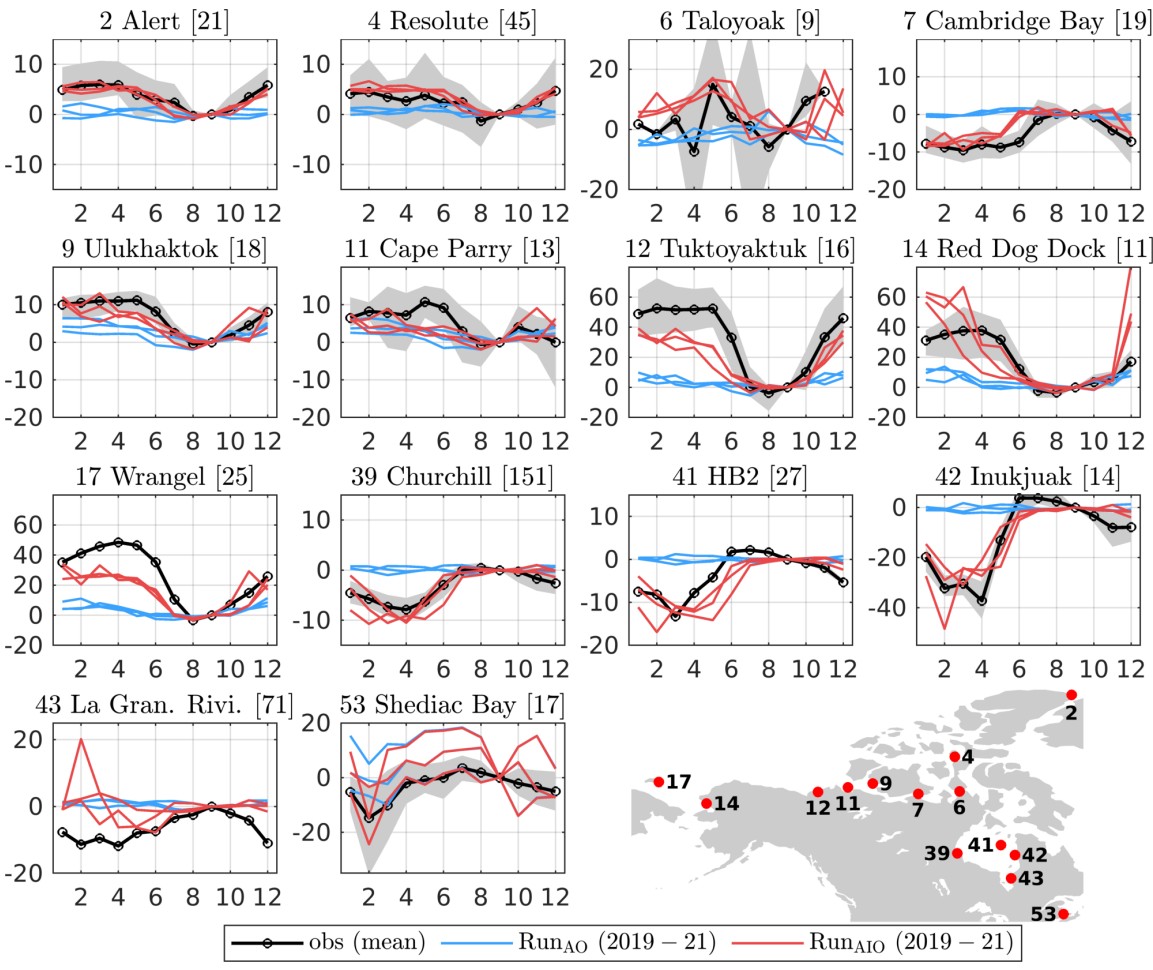

**Figure 7.** Same as Fig. 6 but for the monthly $M_2$ phase anomaly ($\Delta\phi_i$, °) relative to September.

necker and Teubner, 1980; Prinsenberg, 1988; Roos and Schuttelaars, 2011). This behavior applies to both real amphidromes

(i.e., amphidrome over the ocean, marked by "D–G") and virtual amphidromes (i.e., amphidrome over land, marked by "C"

and "H"). We note that real amphidromes may become virtual. It is the case for "D" and "E" in the CAA that shift over land

due to the frictional effects.

These displacements explain the observed and predicted $M_2$ modulations in the CAA and HB shown in Figs. 6 and 7. For

example, the shift of "D" leads to the phase advance at station 7 located to its right side (recall, the relative direction is referred

to the direction facing the amphidrome shift). The shift of "C" causes the phase delay at station 4 located to its left side. The

shift of "E" is found responsible for the overestimated amplitude modulation at station 4, suggesting this shift is overestimated.

In the HB, the shifts of "F" and "G" led to the phase advance in most of the HB. In the southern extension of HB, the shift

of "H" led to a phase delay on its left side where station 43 is located, which counters the phase advance induced by the



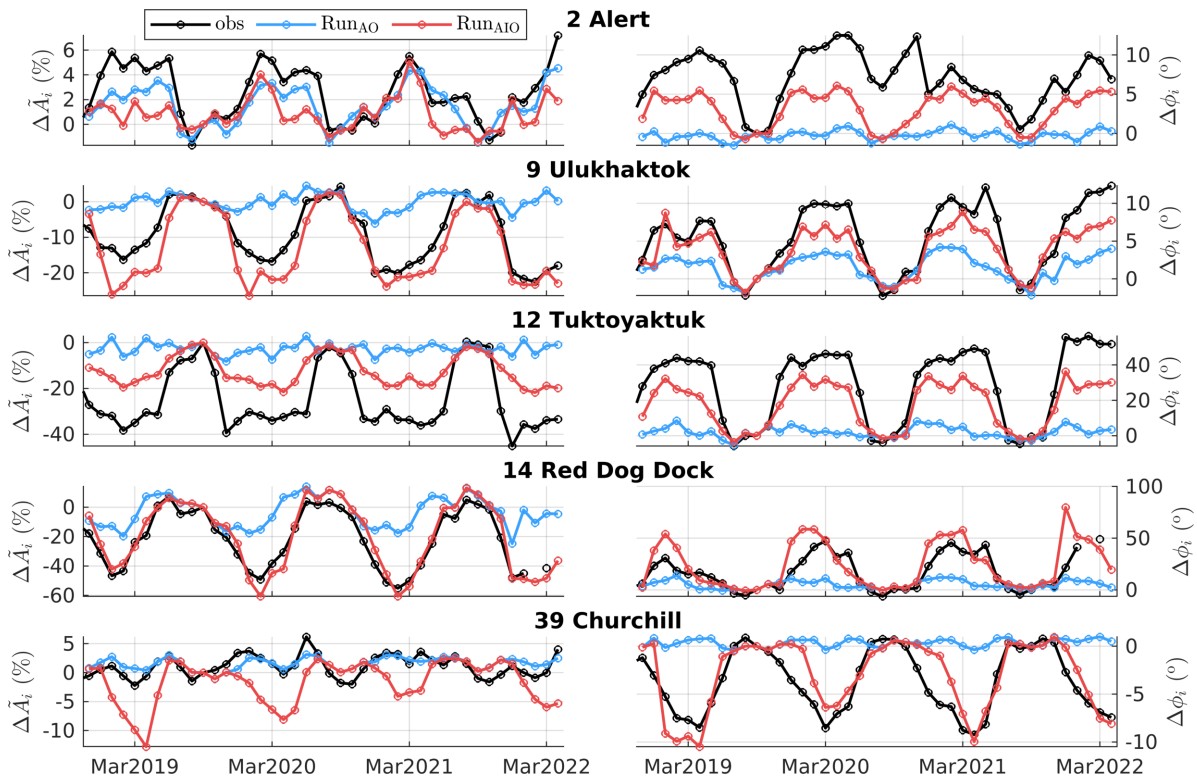

**Figure 8.** Monthly normalized $M_2$ amplitude anomaly ($\Delta \tilde{A}_i$, left panels) and phase anomaly ($\Delta \phi_i$, right panels) relative to September 2019 at five permanent gauges observed (black), Run$_{AO}$ (blue) and Run$_{AIO}$ (red) for the period November 2018 to April 2022.

shifts of "F" and "G". The overall effect is an underestimated phase advance at station 43, indicating that the shift of "H" is

overestimated.

With ice conditions changing rapidly in the Actic, it is important to know where the presence of ice or the under-ice friction is most relevant for the amphidrome shifts shown above. Similar to bottom friction, we expect the impact of the under-ice friction is most relevant over shallow waters where tidal dissipation is significant. We thus conducted four additional sensitivity runs by applying the under-ice friction over regions with water depth less than 50 m, 100 m, 150 m and 200 m, respectively. We

found that applying the friction over water depth less than 100 m can reproduce almost all modulations and the associated amphidrome shifts of Run$_{AIO}$. These important regions (see Fig. 1 for bathymetry), combined with significant presence of landfast or non-free drift ice (see top panels of Fig. 3), cover the bulk of the ESS and CS, and the shallow waters (less than 100 m) of the Canadian Arctic and the HB system. This also indicates that although there are large amounts of landfast ice in the deeper waters (greater than 100 m, top panels of Fig. 3) of the CAA, due primarily to arch formation between islands, their

impact on tides is insignificant.



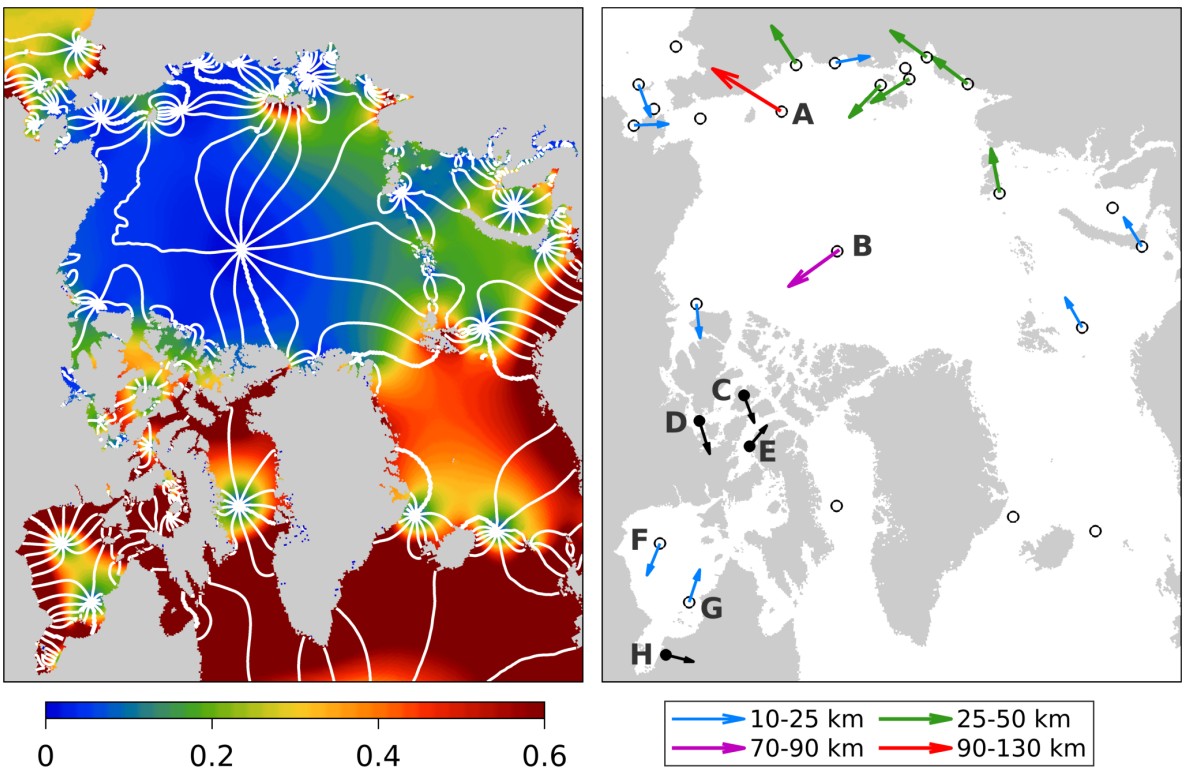

**Figure 9.** Left panel: M$_2$ amphidromes in March predicted by Run$_{AO}$ with amplitude (m) in contour map and phase (every 30°) in white lines. Right panel: Ice-induced displacements of M$_2$ amphidromes in March, taken as the difference in predicted amphidromic points between Run$_{AIO}$ and Run$_{AO}$. The unfilled circles denote real amphidromes. Their displacements, averaged for 2019-2022, are denoted by colored arrows. The filled circles denote virtual amphidromes or amphidromes that become virtual after the displacement. Their positions and displacements (denoted by black arrows) are for illustration purpose only (not to scale).

### 6.2.3 Impact of tide-surge interaction and baroclinicity

We next examine individual contributions from TSI, baroclinicity and under-ice friction to $\Delta \tilde{A}_{\mathrm{Mar}}$ and $\Delta \phi_{\mathrm{Mar}}$ for M$_2$ (Fig. 10) based on process-oriented runs (Runs 1–4, Table 2). As expected, the under-ice friction (left panels) has the largest influence while the effect of TSI due to bottom stress is negligible (middle-left panels) due to weak bottom currents. The effects of TSI

due to $\boldsymbol{\tau}_{\mathrm{io}}$ (middle-right panels) and the effects of baroclinicity (right panels) are non-negligible. The two mechanisms act in different ways, leading to different spatial and temporal signatures. Strong TSI due to $\boldsymbol{\tau}_{\mathrm{io}}$ occurs predominantly over the ESS and CS in March, due to the combination of large ice cover and strong wind-driven surface currents induced by frequent winter storms. Its main effect is to reduce the local M$_2$ amplitude (>10%), resulting in shifts of many small to moderate local amphidromes. It also drives a phase delay of 10–40° over most of the ESS, CS, and Beaufort Sea. Overall, over most of the

affected areas, this mechanism reinforces the modulations induced by the under-ice friction (compare left and middle-right panels).





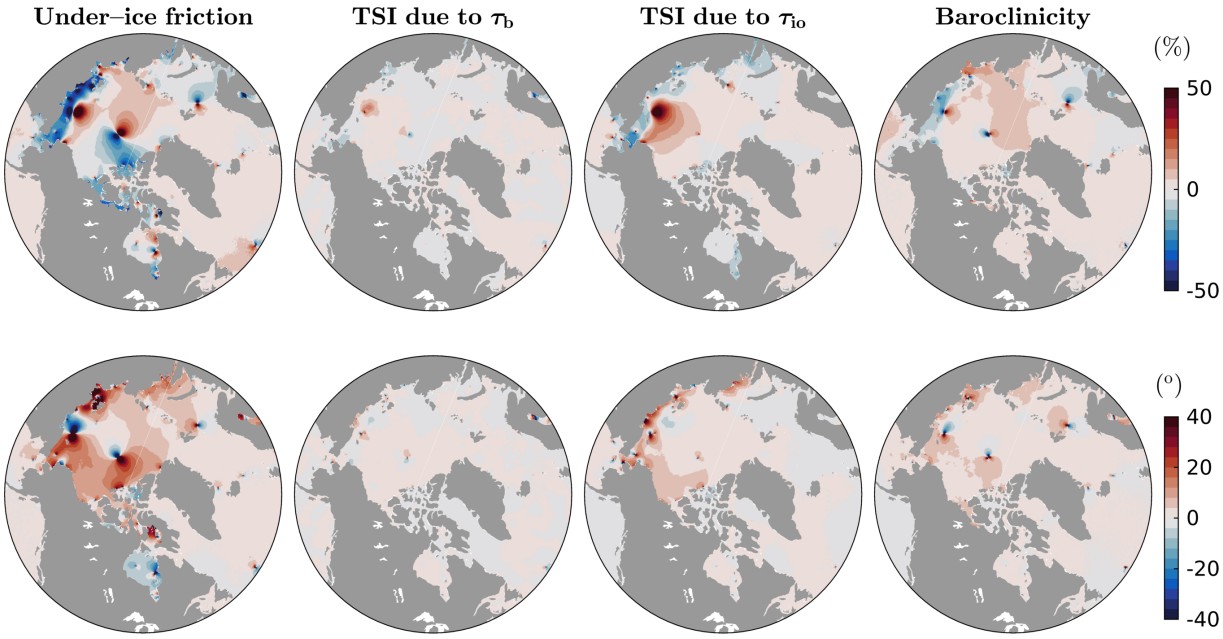

**Figure 10.** Differences in $\Delta \tilde{A}_{\mathrm{Mar}}$ (top panels) and $\Delta \phi_{\mathrm{Mar}}$ (bottom panels) for $M_2$ tides between process-oriented runs: Run3-Run4 (left), Run2-Run3 (middle-left) and Run1-Run2 (middle-right), and $\mathrm{Run_{AIO}}$-Run1 (right), corresponding to effects of the under-ice friction, TSI due to $\boldsymbol{\tau}_b$, TSI due to $\boldsymbol{\tau}_{\mathrm{io}}$, and baroclinicity.

In contrast, we found that modulations induced by baroclinicity occur mainly in September. This is consistent with Müller et al. (2014) who found that the annual maximum tide occurs in summer in the western Yellow Sea and North Sea. This can be explained by baroclinic effects on the vertical profile of eddy viscosity (Müller et al., 2014): the presence of the pycnocline leads to a stabilized water column, and thus reduced tidal dissipation through turbulent processes. Baroclinic effects appear to have a larger scale, mainly affecting several relatively large amphidromes in the Arctic (right panels of Fig. 10). In September, relative to March, this mechanism leads to increased amplitudes (up to 10%) in the ESS, CS, and north of Norway. The corresponding amphidrome shifts are responsible for most of the phase modulations (about $10°$). Compared to left panels of Fig. 10, baroclinicity also reinforces the modulations induced by the under-ice friction.

## 6.3 Storm surges

Storm surges are primarily driven by surface winds and air pressure. Typically when we evaluate the surge component ($\eta_S$) of observed water levels ($\eta_{obs}$), we assume that,

$$\eta_{obs} = \eta_T + \eta_S + \eta_\epsilon \tag{17}$$





where $\eta_T$ is the tide and $\eta_\epsilon$ includes contributions from other processes such as seiches. We further decompose surges into a
wind and pressure driven component so that,

$$\eta_S \approx \eta_W + \eta_P \qquad (18)$$

where $\eta_W$ is the isostatic wind adjustment part of $\eta_S$, and $\eta_P$ is the inverse barometer effect. A significant contribution to the
variability of surges comes from $\eta_P$, and this part of the surge signal is almost unaffected by $\boldsymbol{\tau}_{\mathrm{io}}$. This causes difficulties in
visualizing and analysing the ice effects. For this reason, in this section we remove $\eta_P$ from the surge level ($\eta_W = \eta_S - \eta_P$).
To obtain $\eta_P$, we produced an inverse barometer only prediction (i.e. a run driven with surface air pressures only). We then
removed the predicted $\eta_P$ from both observed and predicted $\eta_S$. Figures 11 and 12 show time series of observed and predicted
$\eta_W$ at five permanent gauges where the ice effects are sufficiently large to affect $\eta_W$. Adding ice effects in Run$_{\mathrm{AIO}}$ significantly
improves the model skill during storm events. For example, it attenuates the peaks in $\eta_W$ by up to 1.0 m at Tuktoyaktuk (station
12) and 0.25 m at Alert (station 2). Low-frequency variability is also improved (e.g., at Alert) due to the persistent presence of
sea ice. However, some over-attenuated peaks in $\eta_W$ are also found, particularly at Prudhoe Bay (station 13); for instance, the
largest negative $\eta_W$ in 2019 and largest positive $\eta_W$ in 2020 (3rd rows of Fig. 11). Further investigation shows that increasing
the ice-ocean drag coefficient even to unrealistically large values does not help. This leads us to speculate that ice velocities
are underestimated by GIOPS, possibly as a result of insufficient ice break-up during strong storms.

To verify this speculation, we use in-situ measurements of ice and surface current velocities collected at an offshore mooring
(station S2 offshore in Hošeková et al., 2021) located only about 50 km west of Prudhoe Bay. The wind stress, ice and current
velocities are primarily zonal and parallel to the coast of Phudhoe Bay (Fig. 13). Prior to March 15, the observed ice was
mainly landfast (i.e., ice velocity close to zero) except during a storm event in mid-January. The currents predicted by GIOPS
agree well with observations and are consistent with reasonable attenuations of $\eta_W$ in Run$_{\mathrm{AIO}}$ (bottom panel). Around March
15, during the passage of another storm, observations indicate an ice break-up event associated with strong ice and current
velocities (up to 0.8 m s$^{-1}$). These observed values are greatly underestimated (about 60%) by GIOPS. After March 15, the
observed ice appears to keep drifting most of the time. This feature was poorly modelled by GIOPS, leading to systematically
underestimated ice and current velocities and results in an over-attenuation of $\eta_W$ in Run$_{\mathrm{AIO}}$ from mid-March to mid-April
(bottom panel).

In contrast, the successful attenuation of $\eta_W$ predicted by Run$_{\mathrm{AIO}}$ at Tuktoyaktuk and Alert throughout February–April
suggests that sea ice over the two regions has much stronger resistance to large storms. To further illustrate the impact of ice,
we examine the ice-induced changes in $\boldsymbol{\tau}_s$ during two large storm events: March 15, 2020 at Tuktoyaktuk and April 02, 2020 at
Alert (Fig. 14). In both cases, winds blow parallel to the coast (left panels), generating the 'Ekman setup' associated with large
$\eta_W$ predicted by Run$_{\mathrm{AO}}$, up to 1.3 m at Tuktoyaktuk and 0.4 m at Alert. In Run$_{\mathrm{AIO}}$, sea ice associated with strong internal stress
greatly reduces $\boldsymbol{\tau}_s$. Reductions in $\boldsymbol{\tau}_s$ in the Tuktoyaktuk region are concentrated closer to the coast, and in particular $\boldsymbol{\tau}_s$ is
completely shut down at the coast by landfast ice. Reductions in $\boldsymbol{\tau}_s$ in the Alert region reach 70–80% for the entire storm. This
leads to significant $\eta_W$ attenuation spanning about 1000 km along the coast for each storm (right panels), including the western
Canadian Arctic (up to 1.0 m attenuation), northeastern Canadian Arctic and North Greenland (up to 0.5 m attenuation).



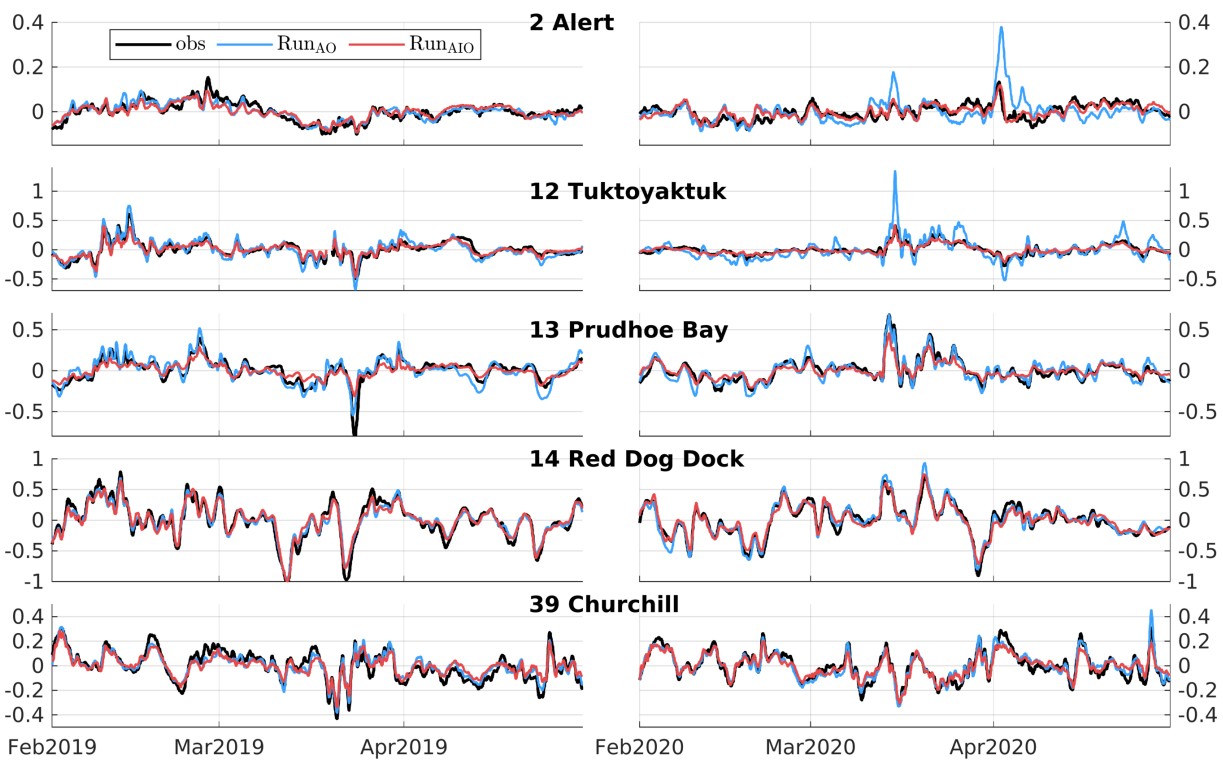

**Figure 11.** Time series of $\eta_W$ observed (black) and predicted by $\text{Run}_{AO}$ (blue) and $\text{Run}_{AIO}$ (red) at five permanent gauges during Feb-Apr of 2019-20. All values in m.

Finally, we note that the comparison of results from $\text{Run}_{AO}$ and $\text{Run}_{AIO}$ also shows large attenuations of $\eta_W$ up to 1.0 m in the Russian Arctic. Although observations are not available, the estimation is expected to be reasonable considering the large amount of landfast ice in the ESS (see Fig. 3).

## 7    Summary and Conclusions

The present study outlines, and evaluates, a novel approach for adding sea ice effects to a global TWL forecast model. Two overriding questions are addressed: (1) Can we design an efficient parameterization to include ice effects in a global ocean model for forecasting TWL and improve forecast skill in polar regions? (2) Can we isolate and explain the contribution of dominant physical processes (e.g., under-ice friction, baroclinicity, nonlinear tide-surge interaction) to the seasonal modulation of tides?

The approach incorporates the total (tide+surge) ice-induced ocean stress by taking advantage of an already available external forecast fields (i.e., ice concentration, ice velocity and surface ocean currents) produced by operational ice-ocean systems. The new method's novel feature is a consistent representation of the tidal relative ice-ocean velocity based on a transfer function derived from ice and ocean tidal ellipses given by external ice-ocean models. This effectively helps circumvent inconsistencies

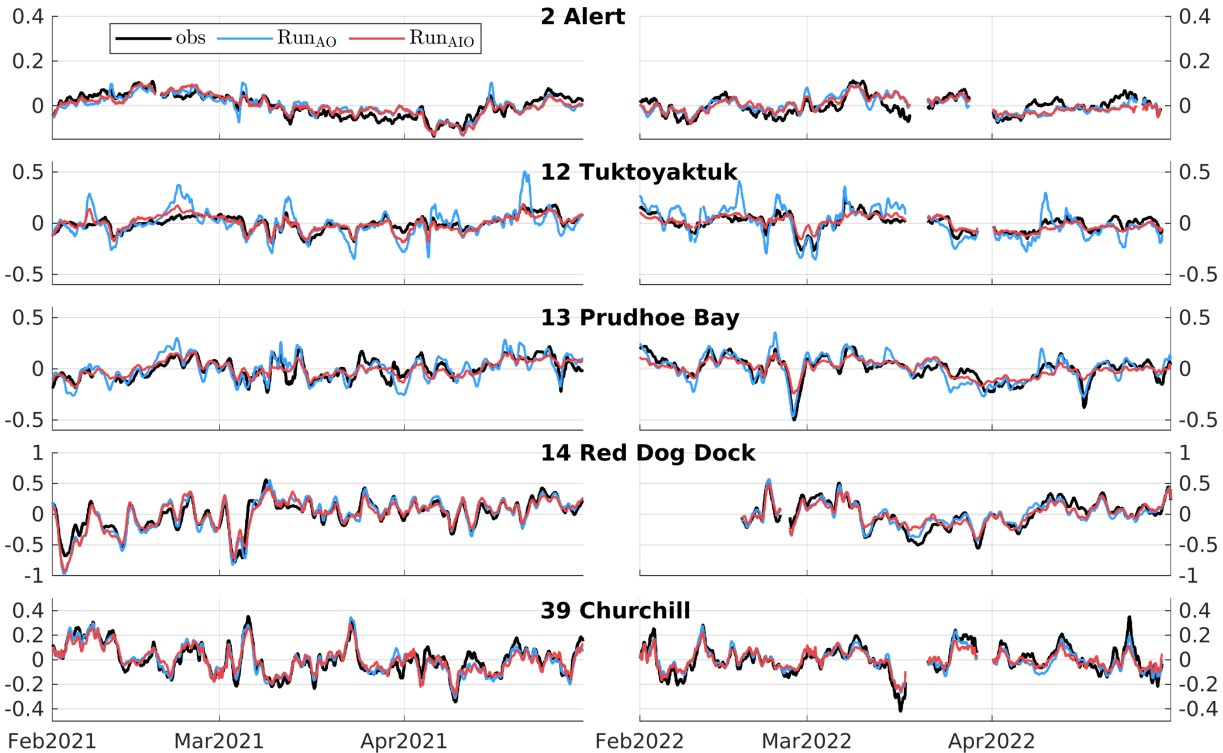

**Figure 12.** Same as Fig. 11 but for 2021-22.

in tides among different models. The approach was applied to ECCC's high resolution (1/12°) global operational TWL forecast system. The external model is a coarser resolution (1/4°), data-assimilative global ice-ocean prediction system also running operationally at ECCC. Model predictions of TWL were generated for the period November 2018 to April 2022 covering four ice seasons.

The impact of adding ice effects was quantified using observed hourly sea level at 58 tide gauges and moorings in ice-infested waters of the Northern Hemisphere. Adding ice effects is shown to help reproduce most of the observed seasonal modulations in the dominant $M_2$ tide (up to 40% in amplitude and 50° in phase) in the CS, Canadian Arctic and HB. The observed interannual variability of the modulation (up to 10% in amplitude and 10° in phase) during the four ice seasons is also captured by the model with the addition of ice effects. Improvements are also found, mostly in the Arctic, for other smaller

constituents (i.e., $S_2$, $K_1$, and $O_1$, see Supplementary material).

The dominant mechanism for seasonal modulations is the under-ice friction due to the presence of landfast or non-free drift ice. It is mostly relevant in areas of shallow waters (less than 100 m) with strong tidal dissipation. These areas cover most of the ESS and CS, and parts of the Canadian Arctic and HB system. The under-ice friction leads to amplitude reductions and phase delays. In turn they can drive large displacements of amphidromes (up to 125 km), resulting in opposite responses

(i.e., amplitude enhancement and phase advance). Remote effects of over-predicted displacements help explain some of the



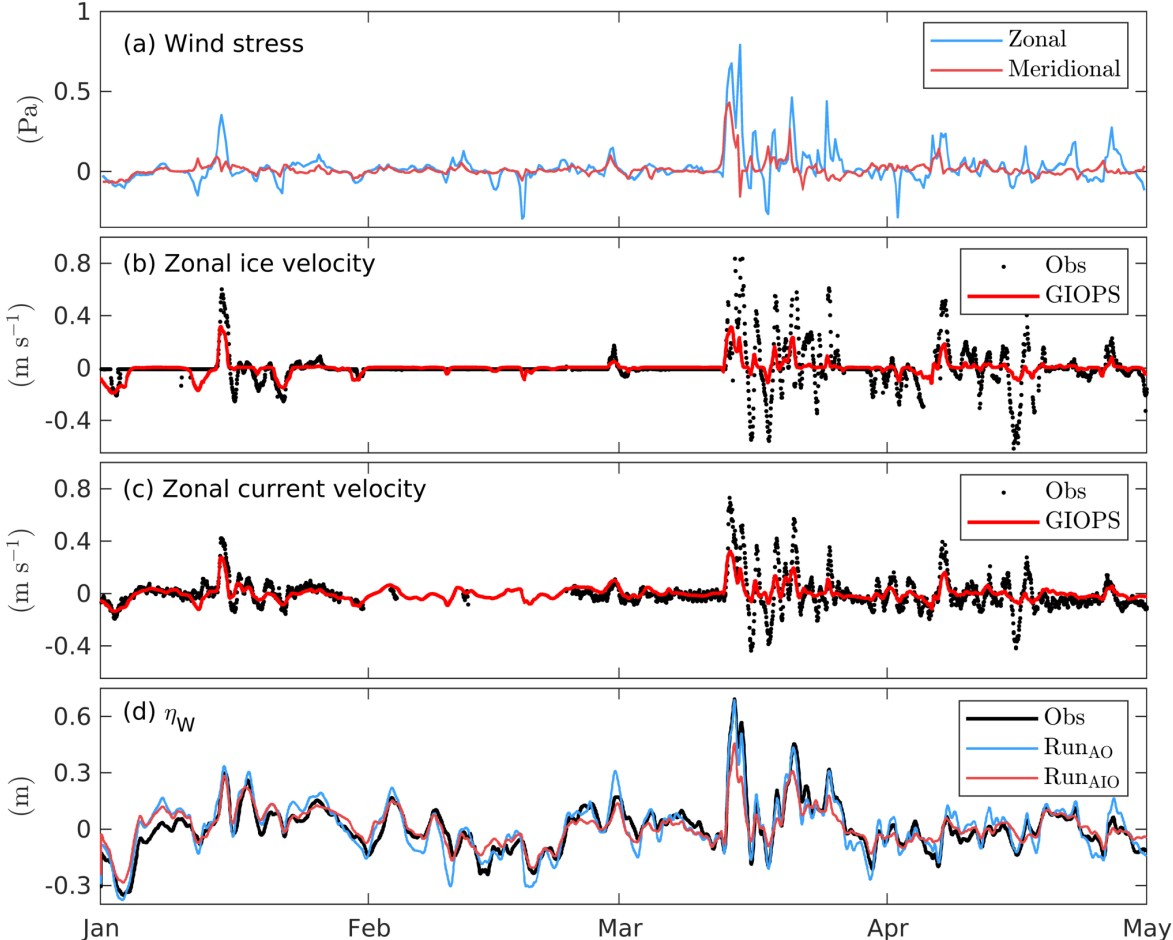

**Figure 13.** Time series of (a) wind stress provided by the GDPS, (b and c) observed and predicted zonal ice and surface current velocities at an offshore mooring near Jones Islands, located about 50 km west of Prudhoe Bay (station 13), from January to April, 2020. (d) Time series of observed and predicted $\eta_W$ at Prudhoe Bay.

discrepancies between observations and model predictions. In addition to the under-ice friction, important contributions from baroclinicity and TSI due to ice-ocean stress were found. The impact of TSI due to ice-ocean stress is found predominantly in March, in shallow areas (i.e., ESS and CS) where there are both large ice cover and strong winter storms. In contrast, baroclinic effects are prominent in September owing to the presence of a pycnocline. Baroclinic mechanism also affect amphidromes, in particular the relatively large amphidromes located in the Arctic Ocean. Both mechanisms, TSI and baroclinic effects, generally reinforce the seasonal modulations induced by under-ice friction.

Adding ice effects also greatly improves the model skill in predicting storm surges. This is achieved via ice-induced surge attenuation (up to 1.0 m) over regions (e.g., western and northeastern Canadian Arctic) associated with weak ice mobility and thus strong ice strength. The attenuation is due to considerable ice-induced reductions (70-100%) of the surface stress. Large



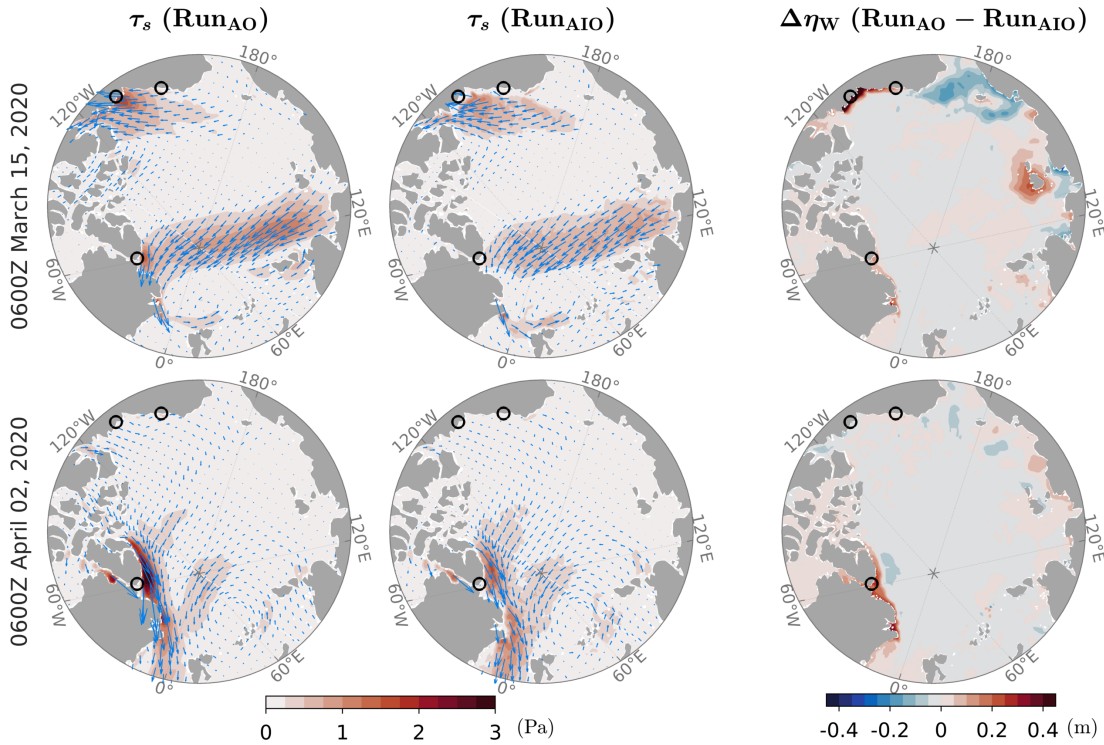

**Figure 14.** Snapshots of surface stress ($\tau_s$) used in Run$_{AO}$ (left) and Run$_{AIO}$ (middle), and differences, $\Delta\eta_W$ (Run$_{AO}$-Run$_{AIO}$), during two storm events in March 15, 2020 (top panels) and April 02, 2020 (bottom panels). Arrows in the left two columns show the wind stress vectors. Circles show the location of three tide gauges (clockwise from the bottom: station 2, Alert; station 12, Tuktoyaktuk; station 13, Prudhoe Bay).

attenuations, up to 1.0 m, are also predicted along the coast of ESS associated with strong ice strength. Forecast challenges remain in regions with intermediate ice strength (i.e. the coast of northern Alaska) where the inclusion of ice effects over-attenuated surges. Insufficient ice break-up during strong storms in the external model was shown to lead to underestimated ice and current velocities (about 60%) compared to in-situ measurements, and they carry across in the form of over-attenuated surges in the TWL system.

Over the next 100 years, climate change is expected to accelerate, causing a general reduction in ice cover in the Arctic (Pörtner et al., 2022). Our results imply that the reduction of ice concentration and strength over areas of shallow waters, in particular, will dramatically increase tidal amplitudes over most coastal areas. As effects of ice and baroclinicity are expected to decrease and increase respectively in winter, tidal amphidromic systems will be pushed towards their ice-free states. The reduced ice cover is also expected to enhance intensification of winter storms (Crawford et al., 2022), contributing to higher storm

surges. Future changes in both tides and storm surges thus pose increasing risk of coastal flooding and erosion, particularly for coastal areas in the Canadian Arctic currently fully or partially protected by the ice cover.



In terms of future work, we plan to extend the present study to the Antarctic once ice cavities in the Ross Sea and Weddell Sea are included in the external ice-ocean model. It will also be interesting to investigate the dynamical mechanisms behind the observed counter-intuitive $M_2$ modulation in the Ross Sea reported by Ray et al. (2021).

*Code availability.* Source code of NEMO v3.6 and its configuration for this study can be accessed at https://doi.org/10.5281/zenodo.7662916. The original code was modified to include the new parameterization of the ice-ocean stress.

*Data availability.* Hourly tide gauge records collected from four institutes are publicly available at MEDS (https://www.meds-sdmm.dfo-mpo.gc.ca/isdm-gdsi/twl-mne/index-eng.htm), NOAA (https://tidesandcurrents.noaa.gov/stations.html?type=Water+Levels), UHSLC (https://uhslc.soest.hawaii.edu/data/), and EMOPnet (https://emodnet.ec.europa.eu/en). Data obtained from two publications, including the monthly
tidal constants in the Russian Arctic and bottom pressure records in the Hudson Bay, will be available via request to the corresponding author. In-situ measurements of ice and surface current velocities collected at an offshore mooring is available at http://hdl.handle.net/1773/47139 (last access: 21/02/2023).

*Author contributions.* Both authors contributed to the conceptualization, methodology, and the writing of the paper. PW coded the parameterization of ice-ocean stress, performed the simulation and evaluation. NBB provided supervision and managed the project.

*Competing interests.* The authors declare that they have no conflict of interest.

*Acknowledgements.* We would like to pay our gratitude and deepest respects to our recently deceased long time friend, colleague, and mentor Dr. Keith R. Thompson. It was a privileged to brainstorm and share the love of our science with him. He made valuable comments and suggestions in the development of the new parameterization. We thank him for that and much more. The authors also thank M. E. Kulikov for providing the monthly tidal constants in the Russian Arctic, and P. St-Laurent for providing the bottom pressure records in the Hudson
Bay. Lastly, the authors thank Mathieu Plante for his detailed comments on an early draft of this paper.





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
