# Peer review of "Adding Sea Ice Effects to A Global Operational Model (NEMO v3.6) for Forecasting Total Water Level: Approach and Impact"

_Geoscientific Model Development, 2023_

## Author Comment (AC1)

**Response to Reviewer #1**

*This very good paper has two main threads: (1) presentation of an efficient way to implement effects from sea ice in an operational ocean model and (2) an improved physical understanding of why so many tide gauges in the Arctic display seasonal perturbations in their tidal harmonics. Both justify publication, so I urge the editor's acceptance.*

*For the main results, shown in Figure 4 (total water level), I was somewhat disappointed to see how little improvement was obtained. But my enthusiasm returned when the tide results (Figure 5 and after) were presented. These results are most encouraging.*

*I have only a few minor comments/suggestions.*

Many thanks for taking the time to review our manuscript and providing such constructive comments. Our responses to your specific comments are given below.

*1: Figure 3 caption. It would be useful to have more information about the source of the landfast frequency. Simply citing the Nat. Snow and Ice Data Center is not very informative.*

Agreed. We have modified the caption and included the correct citation of the data as follows:

"Observed frequency of landfast ice occurrence for December 2020 to March 2021, calculated based on the weekly fast ice extent provided by the U.S. National Ice Center (2020)."

*2: Figure 4 caption (very minor). It might help readers (or browsers) to note that the missing stations are those for which only historical observations are available, and those data do not overlap with the model timeframe.*

Following your suggestion, we have added the following text to the caption:

"The missing stations are those for which only historical observations are available, i.e., data do not overlap with the study period."

*3: Line 19: I think a better and more comprehensive reference for Arctic Sea ice changes is a recent paper by Parkinson: https://doi.org/10.3389/frsen.2022.1021781*

Thank you. We have added this good reference (Parkinson, 2022).

*4: The Ray (2022) paper was published by Ocean Sciences: http://dx.doi.org/10.5194/os-18-1073-2022*

Corrected.

*5: For Eqns (1-4), if you are using LaTeX, you can obtain better sized parentheses and brackets by using \left and \right. See the manual.*

Corrected. It looks much better. Thanks.

*6: The discussion in the Supplement of poor results at Nome, Alaska, is interesting, in part because NOAA has noted its water level predictions for Nome are the worst of any U.S. tide gauge. See: https://tidesandcurrents.noaa.gov/pdf/Tide_Prediction_Error_for_the_United_States_Coastline.pdf.*

*The tides at Nome are small, and non-tidal variability is large, according to a spectrum of tidal residuals (http://dx.doi.org/10.1357/002224017821836761, Figure 3), but that does not explain why O1, K1 display such large annual perturbations while the semidiurnal tides do not. A mystery, although it could be related to discharge, as the authors speculate.*

This is a very good point, and we realize that the nonlinear tide-surge interaction could also be an important contributor to the large observed modulations of O1, K1 at Nome. It is also interesting that diurnal tides display large annual perturbations while semidiurnal tides do not. We modified the Supplement to include a brief discussion:

"Egbert and Ray (2017) showed that the non-tidal variability at Nome is large compared to tides, implying potential effects of the nonlinear tide-surge interaction (TSI). We speculate that the observed large modulations of O1, K1 are affected by both sea ice and the TSI. Both are not well captured locally (the model underestimates the amplitudes of O1 and K1, by up to half in ice-free months). It is also interesting to note that in contrast, the semidiurnal tides do not display large modulations. This may be attributed in part to the more complex semidiurnal amphidrome systems over this region (see the left panel of Fig. 9 in the main text), characterized with smaller wavelength than diurnal tides."

**New references mentioned in the response letter:**

Egbert, G. D., & Ray, R. D.: 'Tidal prediction' in The Sea: The Science of Ocean Prediction. *Journal of Marine Research*, *75*(3), 189-237, 2017.

Parkinson, C. L.: Arctic Sea Ice Coverage from 43 Years of Satellite Passive-Microwave Observations. *Frontiers in Remote Sensing*, 94, 2022.

U.S. National Ice Center: U.S. National Ice Center Arctic and Antarctic Sea Ice Concentration and Climatologies in Gridded Format, Version 1. Boulder, Colorado USA. NSIDC: National Snow and Ice Data Center. https://doi.org/10.7265/46cc-3952. 2020.

---

## Author Comment (AC2)

**Response to Reviewer #2**

*This study demonstrates a methodology to efficiently incorporate sea ice effects into a global operational total water level forecast model and analyzes the impact to tides, surge, and total water levels in the Arctic region, including seasonal modulation and attenuation of water levels.*

*The study and manuscript is of high quality and is a useful contribution to the field and the journal. My comments below are mainly minor ones asking for clarification of methodologies and ideas.*

Many thanks for taking the time to review our manuscript and providing such constructive comments. Our responses to your specific comments are given below. Please note that in this response we present figures that we do not plan to include in the main text. These figures are labelled using the form "Figure. S*".

*1. Page 3 L59-60: "However, such parameterizations depend solely on ice concentration, which is insufficient to represent the ice strength and its impact on the air-sea momentum flux transfer". Suggest that the authors either provide reference(s) to support this assertion or more information on the mechanism by which the ice concentration-only parameterization must be insufficient.*

We acknowledge that it cannot be ruled out that the ice concentration-only parameterization could be useful without performing significant additional tests. Nonetheless, it is important to point out limitations. We have modified the text as follows:

"However, one particular challenge with such parameterization is that ice concentration alone cannot fully represent the internal ice stress or the ice strength, which is important for the ice-drift response to winds (e.g., Fissel and Tang, 1991; Heil and Hibler, 2002) and the subsequent ice-ocean momentum transfer. The ice strength is usually a function of both ice concentration and ice thickness (e.g., Heil and Hibler, 2002), for example, higher ice concentration and thicker ice can enhance the ice strength and reduce the ice-ocean momentum transfer."

*2. Page 3 L61-62: "In Canada, sea ice effects on TWL forecasts are a major concern, particularly in the Canadian Arctic and possibly on the east coast of Canada." Why particularly Canadian Arctic and "possibly" the east coast? Is this because that's where the most ice/ice change is or where the winter storms mostly impact? Please provide more detail.*

It is because these regions are where the most ice/ice change is. We have modified the text as follows:

"In Canada, sea ice effects on TWL forecasts are a major concern: sea ice is a prominent feature in the Canadian Arctic and Hudson Bay, and to a much lesser extent on the east coast of Canada."

*3. Page 6, L125, Eq (5): Suggest to add the condition for $\tau_s = \tau_{ao}$, i.e., when ice concentration, $\alpha=0$.*

Added.

*4. Page 7 L136: Any reason why newer versions of GEBCO (2019-2022) bathymetry are not used?*

This is a good point. We didn't want to change the bathymetry mid-way through the model development. However, we plan to examine the newer versions of GEBCO bathymetry in future development and update the model bathymetry if needed.

*5. Page 9 L208-210: The ice-ocean tide rotation angle values are described but not shown. Suggest to add row to figure 3 or add new figure to supplementary for rotation angle, $\phi$.*

Following your suggestion, we've added a 3$^{rd}$ row in Fig. 3 to show the rotation angle, $\phi$.

[Figure]

Figure 3. Top panels: Observed frequency of landfast ice occurrence for December 2020 to March 2021, calculated based on the weekly fast ice extent provided by the U.S. National Ice Center (2020). Middle and bottom panels: Derived monthly $a^T$ and $\phi$ for the $M_2$ tide for the same period. Note that for areas with very weak $u_{ice}^T$ (major axis velocity magnitude less than $5\times10^{-3}$ m s$^{-1}$ in this study), $\phi$ is irrelevant, its estimation is also not reliable, and so it is set to zero.

*6. Page 10 L230: I'm not sure I quite understand what the scale factor, $a^S$ is, how this factor is determined, and why it is a spatial constant. On L182 it says it is used to account for differences in valid depths between the external model and the TWL model. Does this mean the levels at which the model surface currents are valid or the seabed depths? But the text also seems to say it is actually determined from observations (which kinds? Section 2 does not describe any velocity data) not by comparing the models. I think this scale factor needs more and clearer explanation.*

Sorry for the confusion. The valid depth means the levels at which the model surface currents

are valid. The scale factor, $a^s$, is used to adjust the residual relative velocity from the surface level of the external model to that of the TWL model, so a spatial constant is appropriate. In this study, $a^s$ was determined by comparing modelled and observed residual water levels, i.e., it was tuned to best reproduce the observed residuals. We have modified the text as follows:

"Since the tidal and residual relative velocities are calculated based on surface currents coming from the TWL and external models, respectively, their valid surface levels could be different. To be consistent, an empirical scale factor, $a^S$, can be used to adjust the residual relative velocities to the surface levels of the TWL model,

$$\boldsymbol{u}_{\text{ice}}^S - \boldsymbol{u}_{\text{surf}}^S = a^s(\boldsymbol{u}_{\text{ice}}^{S*} - \boldsymbol{u}_{\text{surf}}^{S*})$$

For example, if the surface level is shallower in the external model than the TWL model, $\boldsymbol{u}_{\text{surf}}^{S*}$ will be stronger than $\boldsymbol{u}_{\text{surf}}^S$, and $a^s$ should be larger than unity. In practice, $a^S$ can be tuned to best reproduce the observed residual water level in the presence of ice."

*7. Page 11 L240. It is not clear to me how the following "processes" are removed:*
*Run 2: TSI due to $\tau_{io}$ and Run 3: TSI due to $\tau_b$. I think I can guess that in Run 4, $u_{\text{ice}}^T - u_{\text{surf}}^T$ used in Eq. (9) is set to zero? But how is the tide-surge interaction removed from $\tau_{io}$ and $\tau_b$? Please explain and/or write the mathematical conditions.*

We have added the following explanations including mathematical conditions:

"…four process-oriented runs were also conducted by gradually removing relevant processes from Run$_{\text{AIO}}$: baroclinic effects (by using constant T and S, Run1), TSI due to $\tau_{io}$ (Run2), TSI due to $\tau_b$ (Run3), under-ice friction (by setting $\boldsymbol{u}_{\text{ice}}^T - \boldsymbol{u}_{\text{surf}}^T$ to zero, Run4). Specifically, for Run2, TSI due to $\tau_{io}$ is removed by setting $\tau_{io} = \rho_0 C_{io}(|\boldsymbol{u}_{\text{io}}^T|\boldsymbol{u}_{\text{io}}^T + |\boldsymbol{u}_{\text{io}}^S|\boldsymbol{u}_{\text{io}}^S)$, where $\boldsymbol{u}_{\text{io}}^T$ and $\boldsymbol{u}_{\text{io}}^S$ are respectively the tidal and residual relative ice-ocean velocities given by Eqs. (12) and (13). For Run3, TSI due to $\tau_b$ is removed by setting $\tau_b = \rho_0 C_{db}(|\boldsymbol{u}_{\text{b}}^T|\boldsymbol{u}_{\text{b}}^T + |\boldsymbol{u}_{\text{b}}^S|\boldsymbol{u}_{\text{b}}^S)$, where $\boldsymbol{u}_{\text{b}}^T$ and $\boldsymbol{u}_{\text{b}}^S$ are isolated using the online tidal filter (denoted by the angle brackets, Wang et al., 2021): $\boldsymbol{u}_{\text{b}}^T = \langle\boldsymbol{u}_b\rangle; \boldsymbol{u}_{\text{b}}^S = \boldsymbol{u}_b - \langle\boldsymbol{u}_b\rangle$."

*8. Page 13/14 L295-300. The authors mention the anomalously high positive M2 amplitude change for station 18 was not captured by the Run$_{AIO}$ (actually predicts a negative change), and associates it with displacement of local amphidromes. However, Run$_{AIO}$ captures the phase change very well which would seem to contradict this explanation. More evidence to explain this discrepancy is warranted.*

This can be better understood by considering the impact of the under-ice friction as a combination of two processes: (1) direct frictional effects that result in amplitude reductions

(i.e., negative $\Delta\tilde{A}_{\mathrm{Mar}}$) and phase delay (i.e., positive $\Delta\phi_{\mathrm{Mar}}$), and (2) indirect effects through the amphidrome shift that leads to both positive and negative changes of amplitude and phase. In Fig. S1 (bottom panels), we present a close-up view of $\Delta\tilde{A}_{\mathrm{Mar}}$ and $\Delta\phi_{\mathrm{Mar}}$ predicted by $\mathrm{Run_{AIO}}$ over the area where station 18 is located. For phase changes, the direct frictional effect associated with positive $\Delta\phi_{\mathrm{Mar}}$ dominates. For amplitude changes, both direct and indirect effects have important contributions. At station 18, the only possible scenario for a positive $\Delta\tilde{A}_{\mathrm{Mar}}$ is a shift of local amphidrome away from the coast where station 18 is located. Unfortunately, the predicted shift (black arrows) is roughly parallel to that coast. In the revision, we have modified the text as follows:

"For $\mathrm{M_2}$, one exception is station 18, where observations show an "anomalous" positive $\Delta\tilde{A}_{\mathrm{Mar}}$, while $\mathrm{Run_{AIO}}$ generates a negative $\Delta\tilde{A}_{\mathrm{Mar}}$. The only possible scenario for positive $\Delta\tilde{A}_{\mathrm{Mar}}$ is the indirect effect of friction, via a shift of local amphidrome away from the coast where station 18 is located. Unfortunately, the shift predicted by $\mathrm{Run_{AIO}}$ is roughly parallel to that coast. We note however that $\mathrm{Run_{AIO}}$ reproduces well the observed positive $\Delta\phi_{\mathrm{Mar}}$ at station 18, because the direct frictional effects on $\Delta\phi_{\mathrm{Mar}}$ over this area, is much stronger than the indirect effect associated with the amphidrome shift."

[Figure]

Figure S1. Top panels: same as the middle panels in Fig. 5: observed (circles) and predicted (contour map) modulations of the $\mathrm{M_2}$ amplitude ($\Delta\tilde{A}_{\mathrm{Mar}}$, left panels) and phase ($\Delta\phi_{\mathrm{Mar}}$, right panels) in March relative to September. Bottom panels: close-up view of the area where station 18 is located, denoted by the rectangle in the top panels. The arrows denote the shift of the local amphidrome.

*9. Page 21 L384: How did you remove, $\eta_\epsilon$ "contributions from other process such as seiches" to get the surge, $\eta_S$? Or is $\eta_S$ really just the non-tidal residual here?*

Thanks for pointing it out. We should mention that we applied a low-pass filter to the tidal residuals to attenuate high-frequency (less than 10 h) contributions, such as instrument errors

and seiches, that would interfere with the comparison of surges. To clarify this procedure, we have removed Eq. (17) and modified the text as follows:

"Storm surges are primarily driven by surface winds and air pressure, and they are usually represented as the tidal residuals, the differences between observed water levels and predicted tides. However, tidal residuals contain also high-frequency contributions, such as instrument errors and seiches (see Fig. 5 of Wang et al., 2022), which could interfere with the comparison of surges. To attenuate such high-frequency contributions, we applied a low-pass filter with a cut-off period of 10 h to tidal residuals to obtain $\eta_S$. We further decompose $\eta_S$ into a wind and pressure driven component so that, …"

*10. Page 21 L395-398 and Page 25 L455-459: Any ideas how to adjust for the underestimation of ice and current velocities on the northern Alaska coast due to insufficient ice break-up? The ice concentration-only based parameterization used in Joyce et al. (2019) gives the maximum ice-ocean drag coefficient at 50% ice concentration due to the "interplay between the increased number of ice floe edges and additional sheltering of the atmospheric flow downstream of ice floes with increasing sea ice concentration". Although the ice velocities are underestimated in GIOPS, what about the ice concentrations?*

Thanks for the suggestion. We present in Fig. S2 the ice concentration predicted by GIOPS during this storm event. The arrival of the storm caused some minor ice cracks along the northern Alaska coast associated with decreased ice concentration from 0.99 to about 0.60 (bottom-right three panels). However, the overall ice concentrations are higher than 0.98 for most areas covered by the storm. Similar to underestimated ice velocities, ice concentration is likely overestimated due to insufficient ice break-up. It is thus unlikely that the ice concentration-only parameterization will help in this case. Nonetheless, we note that there are ongoing efforts to improve the ice-ocean model.

[Figure]

Figure S2. Top panels: evolution of winds (wind directions in vectors, and magnitude of wind speed in contour map) on March 13-14, 2020. Bottom panels: ice concentration predicted by GIOPS for the same period.

*11. Technical corrections: Page 12 L265: "constitute" to "constituent".*

Corrected.

**New references mentioned in the response letter:**

Fissel, D. B., and C. L. Tang: Response of sea ice drift to wind forcing on the northeastern Newfoundland shelf. *Journal of Geophysical Research: Oceans* 96.C10: 18397-18409, 1991.

Heil, P., & Hibler, W. D.: Modeling the high-frequency component of Arctic sea ice drift and deformation. *Journal of Physical Oceanography*, *32*(11), 3039-3057, 2002.

U.S. National Ice Center: U.S. National Ice Center Arctic and Antarctic Sea Ice Concentration and Climatologies in Gridded Format, Version 1. Boulder, Colorado USA. NSIDC: National Snow and Ice Data Center. https://doi.org/10.7265/46cc-3952. 2020.